# Fact or Hallucination? An Entropy-Based Framework for Attention-Wise Usable Information in LLMs

## Abstract

Large language models (LLMs) often generate confident yet inaccurate outputs, posing serious risks in safety-critical applications. Existing hallucination detection methods typically rely on final-layer logits or post-hoc textual checks, which can obscure the rich semantic signals encoded across model layers. Thus, we propose **Shapley NEAR** (Norm-basEd Attention-wise usable infoRmation), a principled, entropy-based attribution framework grounded in Shapley values that assigns a confidence score indicating whether an LLM output is hallucinatory. Unlike prior approaches, Shapley NEAR decomposes attention-driven information flow across all layers and heads of the model, where higher scores correspond to lower hallucination risk. It further distinguishes between two hallucination types: *parametric hallucinations*, caused by the model's pre-trained knowledge overriding the context, and *context-induced hallucinations*, where misleading context fragments spuriously reduce uncertainty. To mitigate parametric hallucinations, we introduce a test-time *head clipping* technique that prunes attention heads contributing to overconfident, context-agnostic outputs. Empirical results in four QA benchmarks (CoQA, QuAC, SQuAD, and TriviaQA), using Qwen2.5-3B, LLaMA3.1-8B, and OPT-6.7B, demonstrate that Shapley NEAR outperforms strong baselines, without requiring additional training, prompting, or architectural modifications.

## 1 Introduction

The rapid proliferation of large language models (LLMs) in a variety of applications, from conversational agents to automated decision making systems, has underscored their impressive capabilities Ouyang et al. (2022); OpenAI Achiam et al. (2023). However, a challenge persists: these models often generate outputs that are confidently stated yet factually incorrect, a phenomenon widely known as hallucination Ji et al. (2023). This issue becomes especially critical in safety-sensitive environments where factual accuracy is paramount Cohen et al. (2023); Ren et al. (2022).

To tackle this, a number of recent studies have investigated hallucination in LLMs using both theoretical and empirical approaches. While token-level uncertainty measures such as entropy and confidence have proven useful in hallucination detection for NLP tasks Huang et al. (2023), extending these methods to sentence-level predictions in autoregressive LLMs remains challenging due to the models' complex and interdependent outputs Duan et al. (2023); Kuhn et al. (2023). As a workaround, recent research has attempted to infer sentence-level uncertainty directly from the generated language itself Lin et al. (2023); Zhou et al. (2023). However, these works did not consider the dense semantic information encoded inside the internal layers of the LLM Chen et al. (2025; 2024); Xu et al. (2020). In parallel, Xu et al. (2020) introduced the concept of $\mathcal{V}$-usable information, which quantifies how much useful information a model can extract under computational constraints. Building on this, Ethayarajh et al. (2022) proposed Pointwise V-Information (PVI) to estimate instance-level dataset difficulty, although this metric only considers the final layer. In contrast, Chen et al. (2024) proposed using the EigenScore of the final token from a middle transformer layer to detect hallucinations, and further analyzed model reliability by comparing multiple responses to a shared prompt. However, despite these advances, most of these methods focus exclusively on final-layer logits and overlook the rich information encoded in all the internal states of LLMs Azaria & Mitchell (2023). With further development, LLM-Check Sriramanan et al. (2024b) extended hallucination detection to both

Figure 1: Overview of the proposed pipeline for detecting hallucinations. Shapley NEAR detects hallucination by computing entropy-based information gain across all attention heads and layers, and attributing it fairly to individual context sentences using Shapley values.

white-box and black-box settings by employing an auxiliary LLM to analyze hidden states, attention patterns, and output probabilities. Similarly, Lookback Lens Chuang et al. (2024a) trained a linear classifier using the ratio of attention on the context versus generated tokens to identify contextual hallucinations. However, both approaches fail to distinguish whether hallucinations originate from the pre-trained knowledge of the model (parametric hallucination) or from misleading contextual information (contextual hallucination). Complementing these lines of work, Kim et al. (2024) examined deficiencies across layers for unanswerable question detection, while Chen et al. (2025) revealed that feed-forward layers often exhibit less reliable distributional associations compared to the more robust in-context reasoning encoded by attention mechanisms.

To address the limitations of these prior approaches, we introduce **Shapley NEAR (Norm-basEd Attention-wise usable infoRmation)**, a method designed to assign a confidence score indicating whether an LLM-generated answer is trustworthy or hallucinatory, given a question and context. In contrast to previous methods that primarily rely on outputs from feed-forward layers, which have limited bearing on reasoning Chen et al. (2025), our approach focuses exclusively on attention layers. Shapley NEAR aggregates information from all attention heads across all layers Azaria & Mitchell (2023), enabling a fine-grained, attention-wise and layer-wise analysis of information propagation. Crucially, our method requires no additional training or architectural changes, making it both easy to integrate into existing pre-trained models and highly interpretable in practice. The main contributions of our paper are as follows:

- We propose **Shapley NEAR**, a principled, interpretable entropy-based attribution method grounded in Shapley-value theory that quantifies usable information flow in LLMs by decomposing entropy reduction across layers and heads using the norm of attention outputs.

- We demonstrate that our framework not only detects hallucinations introduced by context segments but also distinguishes between *parametric* and *context-induced* hallucinations.

- We introduce a test-time strategy to identify attention heads that consistently exhibit parametric hallucinations. Selectively removing these heads during inference demonstrates a novel application of attribution techniques to improve model reliability without retraining.

- We evaluate Shapley NEAR on multiple QA datasets using Qwen2.5-3B, LLaMA3.1-8B, and OPT-6.7B, showing that it outperforms strong baselines mention in Section.

## 2 BACKGROUND

In this work, we focus on quantifying how much usable information a generative language model can extract from a given context to answer a specific question. Formally, we consider an input context $X = \{s_1, s_2, \ldots, s_n\}$, and a typical autoregressive large language model (LLM), denoted by $\mathcal{V}$, which generates a response sequence $Y = [y_1, y_2, \ldots, y_T]$, where each token $y_t$ is conditioned on the input and previous outputs. Our central goal is to determine how much $\mathcal{V}$-usable information the model can leverage from the context $X$ to predict the output $Y$. A lower value of usable information implies greater prediction difficulty, indicating that the dataset is more challenging for the models $\mathcal{V}$.

While classical information-theoretic tools such as Shannon's mutual information $I(X;Y)$ Shannon (1948) and the data processing inequality (DPI) Pippenger (1988) have long served as foundational metrics for analyzing information flow, recent research has revealed their limitations when applied to deep models. These classical measures tend to overestimate the practically usable signal, particularly in settings where models operate under computational constraints as modern LLMs can progressively extract structured and meaningful representations from raw inputs through deep computation, rendering traditional metrics insufficient.

To bridge this gap, Xu et al. (2020) introduced the notion of *predictive $\mathcal{V}$-information*, which accounts for the computational limitations of a model family $\mathcal{V}$. They define this as the difference between two entropy terms: the conditional $\mathcal{V}$-entropy with and without contextual input. Specifically, the predictive $\mathcal{V}$-information is given by:

$$I_{\mathcal{V}}(X \to Y) = H_{\mathcal{V}}(Y|\emptyset) - H_{\mathcal{V}}(Y|X),$$

where $H_{\mathcal{V}}(Y|X)$ denotes the expected uncertainty over outputs $Y$ when conditioned on context $X$, and $H_{\mathcal{V}}(Y|\emptyset)$ captures the model's uncertainty in the absence of any input. While predictive $\mathcal{V}$-information captures dataset-level trends, Ethayarajh et al. (2022) extend it to the instance level via *pointwise $\mathcal{V}$-information (PVI)*, which measures how much information a specific input $x$ provides for predicting its output $y$. This enables fine-grained analysis of instance difficulty, essential for real-world model evaluation.

Building on these foundations, Kim et al. (2024) propose *layer-wise usable information* ($\mathcal{L}I$), a method that decomposes usable information across the layers of a model, thereby enhancing interpretability. Complementary to this, Chen et al. (2025) show that feed-forward layers primarily encode superficial distributional patterns, whereas attention mechanisms are more closely aligned with in-context reasoning. These insights motivate our work, which integrates the strengths of previous efforts to develop a unified, interpretable framework to assess usable information in LLMs, both across layers and at the sentence level, while accounting for how different components of the model influence predictive certainty.

## 3 SHAPLEY NEAR: NORM-BASED ATTENTION-WISE USABLE INFORMATION

Given a set of context passages, generative language models (LLMs) produce free-form text responses to questions. In this work, we aim to systematically quantify how individual parts of the context influence the prediction at the final token of the question. Transformer-based models organize computation across multiple layers and attention heads, where each head captures distinct patterns of contextual dependency Wang et al. (2022). Building on this insight, we propose **Shapley NEAR**, a framework for measuring how much usable information each sentence in a context contributes to reducing the model's predictive uncertainty. Shapley NEAR is computed by isolating the output of each attention head at the final token position of the question and measuring the change in entropy when conditioning on subsets of the input context versus a null context. To attribute this entropy reduction fairly to individual sentences, we adopt a Shapley-value-based decomposition. For clarity, the remainder of the paper, we will use the terms *Shapley NEAR* and *NEAR* interchangeably. An overview of our architecture is illustrated in Figure 1, while the detailed algorithmic procedure is presented in Appendix A7.

Let $s_x = (s_1, s_2, \ldots, s_n) \in C$ denote a context passage composed of $n$ disjoint sentences, and let $q \in Q$ represent the associated question. The concatenated input sequence $s_x q$ is tokenized into a sequence of length $T$, with the final token of the question indexed by $q_t \in \{1, \ldots, T\}$. In this framework, we consider a formally defined predictive family $\mathcal{V}$ consisting of pretrained generative language models, where each model is composed of $L$ transformer layers and each layer contains $H$ attention heads. Each attention head $h$ in each layer $\ell$ of the language models creates different computations. Mathematically, we define $\mathcal{V} \subseteq \Omega = \{f^{(l,h)} : C \cup \emptyset \to \mathcal{P}(\mathcal{Q})\}$, where $C$ and $Q$ are random variables with sample spaces $\mathcal{C}$ and $\mathcal{Q}$, respectively, and $\mathcal{P}(\mathcal{Q})$ denotes the set of all probability measures over $\mathcal{Q}$ equipped with the Borel algebra on $\mathcal{C}$. The mapping $f^{(l,h)}$ represents the function associated with attention head of a specific layer $(l, h)$ within the predictive family $\mathcal{V}$. The range of $f$ corresponds to the vocabulary space of the model. Given a layer $l$ and attention-head $h$ in $\mathcal{V}$, the function $f$ maps the context tokens (or null context) to probability distribution over the vocabulary. Unlike prior work, the function $f$ is assumed to operate without any additional fine-tuning

on external training data. In the rest of the section we will build the mathematical formula for NEAR, defining and explaining each step.

**Definition 3.1** (Norm-based Attention Information). Prior research by Kobayashi et al. (2020) suggests that the norm of the attention output serves as a meaningful proxy for the amount of information transmitted by each head. We omit the output of the feedforward layers (FC), as previous work by Chen et al. (2025) has shown that these layers predominantly capture shallow distributional associations, whereas the attention layers are more effectively engaged in in-context reasoning.

For each layer $\ell \in \{1, \dots, L\}$ and head $h \in \{1, \dots, H\}$, given an input context subset $x$ and a question $q$, we compute the attention output of the model $\mathcal{V}$ for the combined input $(x, q)$ as follows:

$$\alpha^{(\ell,h)}(x, q) \triangleq \text{softmax}\left(\frac{Q^{(\ell,h)}(x, q)\, K^{(\ell,h)}(x, q)^\top}{\sqrt{d}}\right),$$

$$Z^{(\ell,h)}(x, q) \triangleq \alpha^{(\ell,h)}(x, q) V^{(\ell,h)}(x, q), \tag{1}$$

where $Q^{(\ell,h)}$ and $K^{(\ell,h)}$ denote the query and key matrices for layer $\ell$ and head $h$, respectively, $\alpha^{(\ell,h)}(x, q) \in \mathbb{R}^{T \times T}$ and $V^{(\ell,h)}(x, q) \in \mathbb{R}^{T \times d}$ are the value matrices with $d = D/H$ being the per-head dimension. Both attention weights and value vectors are computed based on the concatenated subset $x$ and question $q$. The resulting attention outputs are projected using equation 1 and a head-specific output matrix $W_O^{(h)} \in \mathbb{R}^{d \times D}$ to obtain

$$\tilde{Z}^{(\ell,h)}(x, q) \triangleq Z^{(\ell,h)}(x, q) W_O^{(h)} \in \mathbb{R}^{T \times D}. \tag{2}$$

According to Azaria & Mitchell (2023); Ren et al. (2022), the last token embedding captures the semantic information of the entire text. Therefore, we then extract the projected vector corresponding to the final question token $q_t$ from equation 2,

$$\mathbf{z}_{x,q}^{(\ell,h)} \triangleq \tilde{Z}_{q_t}^{(\ell,h)} \in \mathbb{R}^D,$$

which serves as a summary of information flow from the context subset $x$ towards predicting the next token after the question. Now we will define the information gain from $x$ for a specific head.

**Definition 3.2** (Information Gain). From Definition 3.1, the vector $\mathbf{z}_{x,q}^{(\ell,h)}$ encapsulates dense semantic information preserved within the internal attention mechanisms of LLMs. By applying a softmax operation over $\mathbf{z}_{x,q}^{(\ell,h)}$, we obtain a vocabulary distribution $\mathbf{p}_{x,q}^{(\ell,h)} \in \mathbb{R}^{|V|}$. The entropy at the final token is computed as

$$\mathcal{H}^{(\ell,h)}(q_t \mid q_{<t}, x) \triangleq -\sum_{i=1}^{|V|} p_i^{(\ell,h)} \log p_i^{(\ell,h)}. \tag{3}$$

We emphasize that entropy is calculated over the entire softmax-normalized vocabulary. This is a critical distinction: hallucination often stems not from low confidence in the correct token alone, but from broad misallocation of probability mass across incorrect options. Therefore, full entropy measurement enables us to detect whether the model's uncertainty is genuinely reduced when informative context is provided. Now to calculate the information gain provided by the subset $x$ at head $h$ and layer $\ell$, it is defined as the reduction in entropy relative to a null context (i.e., no input) using equation 3,

$$\text{IG}^{(\ell,h)}(x \to q) \triangleq \mathcal{H}^{(\ell,h)}(q_t \mid q_{<t}, \emptyset) - \mathcal{H}^{(\ell,h)}(q_t \mid q_{<t}, x), \tag{4}$$

where $\mathcal{H}^{(\ell,h)}(q_t \mid q_{<t}, \emptyset)$ is computed solely from the model's parametric knowledge, without access to any retrieved context. Summing over all heads and layers yields the total information gain using 4:

$$\text{IG}(x \to q) \triangleq \sum_{\ell=1}^{L} \sum_{h=1}^{H} \text{IG}^{(\ell,h)}(x \to q). \tag{5}$$

The quantity $\text{IG}(x \to q)$ captures the behavior of the function $f^{(\ell,h)} : C \cup \emptyset \to \mathcal{P}(\mathcal{Q})$, which maps a context input, or its absence, to a probability distribution over the vocabulary space $\mathcal{Q}$ for each attention head and layer. Moreover, $\text{IG}(x \to q)$ quantifies the amount of information that the context $x$ provides about the question $q$.

Table 1: **Hallucination detection performance evaluation** across four QA datasets (CoQA, QuAC, SQuAD, TriviaQA) and three LLMs (Qwen2.5-3B, LLaMA3.1-8B, OPT-6.7B). We report average AUROC (AUC), Kendall's $\tau$, and Pearson correlation coefficient (PCC) for various baseline methods. Higher values indicate better performance. NEAR achieves the best overall performance.

| Models | CoQA | | | QuAC | | | SQuAD | | | TriviaQA | | |
|---|---|---|---|---|---|---|---|---|---|---|---|---|
| | $\overline{\text{AUC}}\uparrow$ | $\overline{\tau}\uparrow$ | $\overline{\text{PCC}}\uparrow$ | $\overline{\text{AUC}}\uparrow$ | $\overline{\tau}\uparrow$ | $\overline{\text{PCC}}\uparrow$ | $\overline{\text{AUC}}\uparrow$ | $\overline{\tau}\uparrow$ | $\overline{\text{PCC}}\uparrow$ | $\overline{\text{AUC}}\uparrow$ | $\overline{\tau}\uparrow$ | $\overline{\text{PCC}}\uparrow$ |
| **Qwen2.5-3B** | | | | | | | | | | | | |
| P(True) | 0.48 | 0.32 | 0.30 | 0.49 | 0.33 | 0.31 | 0.51 | 0.34 | 0.32 | 0.50 | 0.33 | 0.31 |
| Pointwise $\mathcal{V}$I | 0.51 | 0.35 | 0.32 | 0.50 | 0.34 | 0.31 | 0.52 | 0.36 | 0.33 | 0.53 | 0.36 | 0.34 |
| Usable $\mathcal{L}$I | 0.67 | 0.45 | 0.41 | 0.66 | 0.44 | 0.40 | 0.68 | 0.45 | 0.42 | 0.64 | 0.43 | 0.40 |
| Semantic Entropy | 0.70 | 0.47 | 0.44 | 0.68 | 0.45 | 0.42 | 0.69 | 0.44 | 0.41 | 0.72 | 0.46 | 0.43 |
| Loopback Lens | 0.71 | 0.48 | 0.45 | 0.69 | 0.46 | 0.43 | 0.70 | 0.45 | 0.42 | 0.73 | 0.46 | 0.44 |
| INSIDE | 0.76 | 0.54 | 0.49 | 0.75 | 0.53 | 0.48 | 0.74 | 0.54 | 0.50 | 0.77 | 0.55 | 0.49 |
| NEAR | **0.85** | **0.65** | **0.64** | **0.84** | **0.66** | **0.65** | **0.86** | **0.67** | **0.66** | **0.85** | **0.66** | **0.65** |
| **LLaMA3.1-8B** | | | | | | | | | | | | |
| P(True) | 0.52 | 0.34 | 0.31 | 0.53 | 0.35 | 0.32 | 0.56 | 0.37 | 0.34 | 0.55 | 0.36 | 0.33 |
| Pointwise $\mathcal{V}$I | 0.56 | 0.36 | 0.34 | 0.52 | 0.32 | 0.31 | 0.55 | 0.37 | 0.33 | 0.68 | 0.46 | 0.40 |
| Usable $\mathcal{L}$I | 0.74 | 0.49 | 0.44 | 0.69 | 0.46 | 0.41 | 0.71 | 0.47 | 0.43 | 0.63 | 0.45 | 0.40 |
| Semantic Entropy | 0.73 | 0.42 | 0.43 | 0.67 | 0.40 | 0.44 | 0.69 | 0.39 | 0.41 | 0.76 | 0.41 | 0.41 |
| Loopback Lens | 0.74 | 0.43 | 0.44 | 0.68 | 0.41 | 0.44 | 0.70 | 0.40 | 0.42 | 0.76 | 0.42 | 0.41 |
| INSIDE | 0.80 | 0.56 | 0.51 | 0.79 | 0.55 | 0.50 | 0.76 | 0.58 | 0.53 | 0.81 | 0.57 | 0.50 |
| NEAR | **0.85** | **0.66** | **0.61** | **0.84** | **0.65** | **0.60** | **0.86** | **0.68** | **0.63** | **0.85** | **0.67** | **0.60** |
| **OPT-6.7B** | | | | | | | | | | | | |
| P(True) | 0.51 | 0.33 | 0.30 | 0.52 | 0.34 | 0.31 | 0.55 | 0.36 | 0.33 | 0.54 | 0.35 | 0.32 |
| Pointwise $\mathcal{V}$I | 0.55 | 0.35 | 0.33 | 0.51 | 0.31 | 0.30 | 0.54 | 0.36 | 0.32 | 0.66 | 0.44 | 0.38 |
| Usable $\mathcal{L}$I | 0.72 | 0.47 | 0.42 | 0.67 | 0.44 | 0.39 | 0.70 | 0.46 | 0.41 | 0.61 | 0.43 | 0.38 |
| Semantic Entropy | 0.71 | 0.41 | 0.42 | 0.65 | 0.39 | 0.43 | 0.68 | 0.38 | 0.40 | 0.74 | 0.40 | 0.40 |
| Loopback Lens | 0.72 | 0.42 | 0.43 | 0.66 | 0.40 | 0.44 | 0.69 | 0.39 | 0.41 | 0.75 | 0.41 | 0.40 |
| INSIDE | 0.78 | 0.54 | 0.49 | 0.77 | 0.52 | 0.48 | 0.74 | 0.56 | 0.51 | 0.79 | 0.55 | 0.48 |
| NEAR | **0.84** | **0.65** | **0.60** | **0.83** | **0.64** | **0.59** | **0.85** | **0.66** | **0.61** | **0.84** | **0.65** | **0.59** |

**Definition 3.3** (Shapley Sentence Attribution). Now, for the context passage $s_x = (s_1, s_2, \ldots, s_n) \in C$ and associated question $q \in Q$, we aim to quantify the individual contribution of each sentence $s_i$ in the context to the model's total information gain. To do this, we use the Shapley value Lundberg & Lee (2017), a concept from cooperative game theory that fairly assigns credit to each element based on its average marginal contribution. Using the total information gain defined in Equation equation 5, the Shapley value for sentence $s_i$ is computed as:

$$\text{Shapley IG}_i \triangleq \sum_{S \subseteq N \setminus \{i\}} \frac{|S|!(n-|S|-1)!}{n!} \left[ \text{IG}(S \cup \{s_i\} \to q) - \text{IG}(S \to q) \right], \quad (6)$$

where $N = \{1, \ldots, n\}$ is the set of all sentence indices in the context. For each subset $S$ of sentences that excludes $s_i$, the term inside the brackets measures the marginal increase in information gain when $s_i$ is added. The prefactor is the standard Shapley coefficient, which ensures that the contributions are averaged fairly over all possible insertion orders of the sentences.

**Definition 3.4** (Sentence-level NEAR Score). The total information that can be gained from the context with respect to the given question is captured by aggregating the contributions of individual sentences. Using the Shapley values from Equation 6, the NEAR score is defined as:

$$\text{Shapley NEAR}(s_x, q) \triangleq \frac{1}{n} \sum_{i=1}^{n} \text{Shapley IG}_i, \quad (7)$$

which reflects average marginal information gain from context sentences in answering the question.

Thus, based on Definitions 3.1 through 3.4, Shapley NEAR 7 offers a fine-grained decomposition of the total information gain, quantifying how much usable information the model extracts from $s_x$ to answer the question $q$. The Information Gain (IG) 3 measures the contribution of each attention head and layer, while the Shapley Information Gain (Shapley IG) 6 further attributes this information to individual sentence segments within the context. A higher NEAR score indicates greater information utility from the context, implying that the generated output is less likely to be hallucinatory.

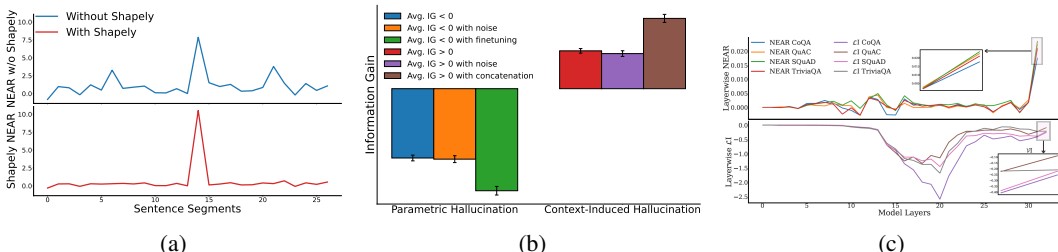

(a)  (b)  (c)

Figure 2: (a) Contribution of Shapley aggregation on an example context where the 14th sentence contains the answer to the question. (b) Information gain scores for detecting parametric and context-induced hallucinations across context segments. (c) Layer-wise information gain comparison between NEAR and $\mathcal{L}I$. As shown in the subgraphs, relying only on the last layer causes loss of information from earlier layers. The last-layer IG of $\mathcal{L}I$ corresponds to $\mathcal{V}I$.

# 4 PROPERTIES AND BOUNDS OF SHAPLEY NEAR

This section outlines the mathematical and experimental properties of NEAR, with derivations in Appendix A1. NEAR aggregates entropy-based information gain across all transformer layers and attention heads, with each term bounded by $\log V$, the maximum entropy over a vocabulary of size $V$. Thus, NEAR is theoretically bounded within $[-L \cdot H \cdot \log V, L \cdot H \cdot \log V]$, where $L$ and $H$ are the number of layers and heads. In practice, it reflects cumulative entropy reduction from contextual conditioning and scales as $\text{NEAR}(s, q) \in O(L \cdot H \cdot \log V)$. Beyond boundedness, NEAR satisfies key behavioral properties. First, it is symmetric: if two context sentences $s_i$ and $s_j$ satisfy

$$\text{IG}(S \cup \{s_i\} \to q) = \text{IG}(S \cup \{s_j\} \to q) \quad \text{for all} \quad S \subseteq s \setminus \{s_i, s_j\},$$

then their Shapley values are identical, i.e., $\text{IG}_i = \text{IG}_j$. We also empirically observed (Section 5) that for each layer $\ell$ and attention head $h$, the following inequality holds:

$$\text{IG}^{(\ell,h)}(\emptyset \to q) \leq \text{IG}^{(\ell,h)}(s_i^{\text{irr}} \to q) \leq \text{IG}^{(\ell,h)}(s_j^{\text{ans}} \to q),$$

here, $s_i^{\text{irr}}$ denotes a context sentence irrelevant to the answer, and $s_j^{\text{ans}}$ contains the ground truth answer. Empirically, NEAR scores also exhibit a monotonicity property similar to information-theoretic measures: for any subset of layers $\mathcal{U} \subseteq L$, the NEAR score computed over $\mathcal{U}$ is always less than or equal to that over the full set $L$, as aggregating more layers cannot reduce total entropy gain:

$$\text{NEAR}_{\mathcal{U}}(s, q) \leq \text{NEAR}_L(s, q),$$

here, $\text{NEAR}_{\mathcal{U}}$ and $\text{NEAR}_L$ denote NEAR scores computed over the subset $\mathcal{U} \subseteq \{1, \ldots, L\}$ and the full set $L$, respectively. This follows from NEAR's additive structure over head-layer pairs, ensuring information accumulates monotonically as more layers are included.

We estimate the NEAR score using an **Average Marginal Effect (AME)** estimator, which models total information gain as a linear combination of sentence-level contributions. For each query, we sample $M$ random subsets of the $n$ context sentences, compute the corresponding IG values, and fit an $\ell_1$-regularized linear model to recover a sparse attribution vector. Under standard conditions, including $k$-sparsity, sub-Gaussian noise, and a restricted eigenvalue condition, the resulting estimate satisfies, with probability at least $1 - \delta$:

$$\left| \hat{\text{NEAR}} - \text{NEAR} \right| \leq \frac{C k}{n \kappa^2} (LH \log V) \sqrt{\frac{\log(n/\delta)}{M}},$$

where $L$, $H$, and $V$ denote the number of transformer layers, attention heads, and vocabulary size.

# 5 EXPERIMENTS

## 5.1 EXPERIMENTAL SETUP

We classify unanswerable questions by computing NEAR scores to assess whether the response generated by a model should be trusted in a given context, that is, whether the answer to a question

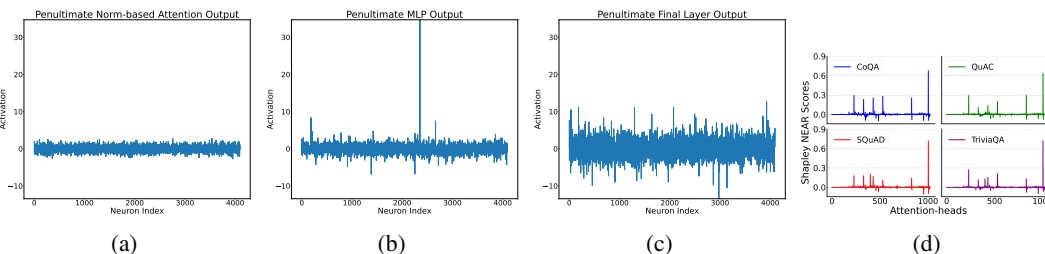

Figure 3: Activation distributions at the final token position in the penultimate layer of LLaMA-3.1-8B: (a) Norm-based attention output, (b) MLP layer output, (c) Final layer output, and (d) Attention-wise Information Gain across all four datasets (CoQA, QuAC, SQuAD, and TriviaQA).

| Methods | AUC ↑ | τ ↑ | PCC ↑ |
|---|---|---|---|
| NEAR w/o Shapley | 0.79 | 0.51 | 0.48 |
| Shapley NEAR | **0.85** | **0.66** | **0.64** |

(a)

| Methods | AUC ↑ | Acc. ↑ | RL ↑ |
|---|---|---|---|
| NEAR | 0.85 | 0.78 | 0.82 |
| INSIDE | 0.80 | 0.74 | 0.80 |
| NEAR + HC | **0.89** | **0.81** | **0.83** |

(b)

Table 2: (a) Contribution of Shapley aggregation to NEAR scores. (b) Head Clipping (HC) results for attention heads with IG $< -0.05$. The following heads were clipped: 349, 459, 485, 833, 955, 1007.

posed can be reliably inferred. We compare NEAR against several strong baselines, including **P(True)** Kadavath et al. (2022), **semantic entropy** Farquhar et al. (2024), **pointwise V-information (PVI)** Ethayarajh et al. (2022), **layer-wise information (LI)** Kim et al. (2024), **Loopback Lens with Sliding Window** Chuang et al. (2024a), and **INSIDE** ($\mathcal{K} = 20$, middle layer of the LLM is considered) Chen et al. (2024). Each method captures a different perspective: P(True) estimates model confidence in binary verification tasks; semantic entropy measures uncertainty via answer diversity; PVI quantifies instance-level predictive difficulty; and LI captures entropy reduction across transformer layers. We evaluate all methods on four question-answering benchmarks: CoQA Reddy et al. (2019), QuAC Choi et al. (2018), SQuAD v2.0 Rajpurkar et al. (2016), and TriviaQA Joshi et al. (2017). Following the setup in Lin et al. (2023), we use the development split of CoQA, validation split of QuAC, a filtered version of the SQuAD v2.0 development set where `is_impossible=True`, and the rc-nocontext validation subset of TriviaQA with duplicates removed. Experiments are conducted on three pretrained models: Qwen2.5-3B, LLaMA3.1-8B, and OPT-6.7B. We report average area under the ROC curve (AUROC), Kendall's $\tau$, and Pearson correlation coefficient (PCC), computed across three independent runs. NEAR scores are estimated with an AME estimator using $M = 50$ randomly sampled coalitions of context sentences and failure probability $\delta = 0.01$, yielding high-confidence estimates of each sentence's contribution to information gain (Appendix A8; further details in Appendix A3). This approximation provides a practical trade-off between computational cost and estimation accuracy, with all reported results exhibiting standard deviations within $\pm 0.04$. We further evaluate on larger models—**LLaMA-3.1-70B** and **Phi-3-Medium-14B**—and on the **LongRA** dataset, comparing NEAR (AME) against **ANAH-v2** and **MIND**. Detailed setup, metrics, and results are provided in Appendix A6.

## 5.2 RESULTS

Table 1 shows the results of hallucination detection using NEAR and several baseline methods across four QA datasets (CoQA, QuAC, SQuAD, and TriviaQA) and three language models (Qwen2.5-3B, LLaMA3.1-8B, and OPT-6.7B). We report performance using AUROC, Kendall's $\tau$, and Pearson correlation (PCC). NEAR consistently performs the best across all datasets and models, showing clear improvements over existing methods. In many cases, it outperforms the strongest baseline, INSIDE, by 8–13% in AUROC and by 10–15% in correlation metrics like $\tau$ and PCC. The best

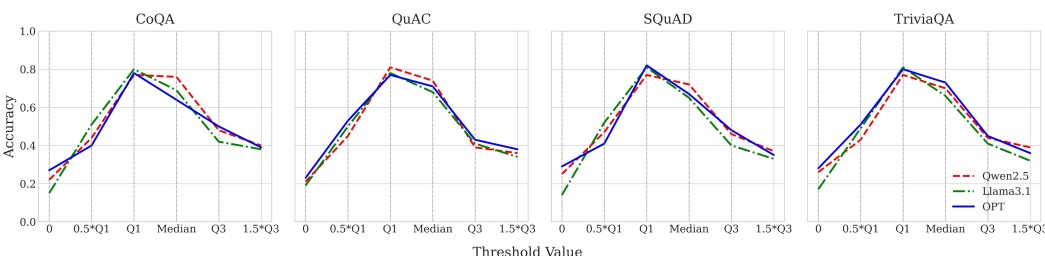

Figure 4: Accuracy vs. NEAR threshold on CoQA, QuAC, SQuAD, and TriviaQA. Optimal separation consistently occurs near the first quartile (Q1) across model variants.

scores for NEAR are observed on the SQuAD dataset for all models, suggesting that SQuAD is easier for LLMs to understand and answer accurately. Among the three models, LLaMA3.1-8B achieves the highest overall performance, ahead of Qwen2.5-3B and OPT-6.7B, especially when used with NEAR. This suggests that stronger pre-trained models can lead to better hallucination detection when combined with effective methods like NEAR. We also evaluated the methods after fine-tuning on the dataset; the results are presented in Appendix A4 and quantitative examples without finetuning in Appendix A9. We also tested NEAR on generalized tasks, detailed in Appendix A6.

## 6 ABLATION STUDIES

For the ablation studies, we primarily focus on the LLaMA-3.1-8B model with the CoQA dataset. Results for other models and datasets are provided in Appendix A2.

**Do we really need to consider all layers instead of only the final layer?** Unlike methods such as $\mathcal{V}$I Ethayarajh et al. (2022), which consider only final-layer outputs, our results show that important semantic information is also captured in earlier layers. As illustrated in Figure 2c, both $\mathcal{L}I$ and NEAR scores indicate that usable information accumulates progressively across inner layers. A similar trend is visible in Figure 3d, where different attention heads capture varying amounts of information. This suggests that focusing only on the final layer overlooks valuable signals present throughout the model.

**Why not consider the output from the layers, as in $\mathcal{L}I$, for NEAR?** Figures 3a and 3b show the activations of the self-attention and MLP components from the penultimate layer of the LLaMA 3.1-8B model. The sharp spikes in these plots reflect extreme internal features in the network, which can cause the model to produce highly overconfident answers Chen et al. (2024); Sun et al. (2021). A similar pattern of overconfidence is also clearly visible in the layer output shown in Figure 3c. We observed this behavior consistently across nearly all layers and LLMs, aligning with the findings of Chen et al. (2025). Based on this evidence, we choose to focus on norm-based attention outputs rather than raw layer activations.

**Detection of Parametric and Context-Induced Hallucinations from NEAR Scores.** Let $s_i \notin \mathcal{A}(q)$ be a context sentence that does not contain the correct answer to question $q$, where $\mathcal{A}(q)$ denotes the set of answer-containing sentences. Ideally, such a sentence should contribute no useful information, and the information gain under attention head $(\ell, h)$ should satisfy $\text{IG}^{(\ell,h)}(s_i \to q) \approx 0$. This follows from equation 4, which becomes negligible when conditioning on $s_i$ does not reduce uncertainty, i.e., $\mathcal{H}^{(\ell,h)}(q_t \mid q_{<t}, s_i) \approx \mathcal{H}^{(\ell,h)}(q_t \mid q_{<t}, \emptyset)$. However, we find that even when $s_i \notin \mathcal{A}(q)$, NEAR scores can be negative ($\text{IG}_i < 0$) or positive ($\text{IG}_i > 0$). A negative score indicates the model becomes more uncertain when conditioned on $s_i$, meaning the context harms rather than helps, this is *parametric hallucination*. A positive score, despite the absence of the answer, implies that the context falsely boosts confidence, this is *context-induced hallucination*. Such cases arise due to in-context learning, the model interprets partial or stylistically similar information as relevant, leading to reduced entropy and overconfidence. To validate this, we measured the mean negative NEAR scores across all context pieces. Adding random noisy text (similar technique used in Chen et al. (2025)) caused negligible change, suggesting that the observed negativity is not due to noise or formulation errors. However, fine-tuning the model on CoQA significantly increased negative NEAR scores, indicating that the model had learned to rely more on context, which led to greater uncertainty

when misleading context was introduced, confirming parametric hallucination. For context-induced hallucination, we computed mean positive NEAR scores for non-answer sentences. While adding random noise had little effect, appending misleading but partially aligned segments of the rest of the context led to a sharp increase in NEAR scores. This confirms that NEAR effectively captures how misleading context increases confidence in incorrect predictions. The results are shown in Figure 2b. However, these hallucinations do not significantly affect the overall reliability of Shapley NEAR, as demonstrated in Appendix A5.

**What Should Be the Threshold Value for NEAR to Segregate Hallucinated Answers?** A key step in using NEAR for hallucination detection is choosing an effective threshold to separate answerable from hallucinated responses. We evaluate classification accuracy by sweeping thresholds across quantiles: $0$, $0.5 \times Q_1$, $Q_1$, Median, $Q_3$, and $1.5 \times Q_3$. As shown in Figure 4, the first quartile ($Q_1$) consistently yields the best accuracy across models (LLaMA-3.1-8B, OPT-6.7B, Qwen2.5-3B) and datasets (CoQA, QuAC, SQuAD, TriviaQA). In contrast, thresholds near 0 or $1.5 \times Q_3$ reduce performance. Based on this, we use $Q_1$ as the default NEAR threshold for all experiments.

**Effect of Shapley Combination on NEAR.** We evaluated the effect of Shapley aggregation in NEAR, comparing it to a greedy method that ranks sentences by standalone gain (without Shapley attribution). As shown in Table 2a, Shapley improves Kendall's $\tau$ ($0.51 \rightarrow 0.66$), PCC ($0.48 \rightarrow 0.64$), and AUC ($0.79 \rightarrow 0.85$), highlighting the benefit of highlighting the benefit of Shapley aggregation over coalitions for robust attribution. Figure 2a shows Shapley downweights irrelevant segments and upweights answer-relevant ones.

**Clipping Heads showing Parametric Hallucination** To further demonstrate the effectiveness of our framework in identifying hallucination-prone attention heads, we clipped all heads in LLaMA-3.1-8B (on the CoQA dataset) with IG values below half the most negative score. This conservative threshold avoids pruning heads with mildly negative IG, which may still contribute useful information (see Figure 3d). We compared our method to INSIDE (EigenScore + Feature Clipping) with a fixed threshold of 0.5, evaluating AUROC, accuracy, and ROUGE-L (computed between the given and generated answers). For both NEAR and NEAR+HC (Head Clipping), we used the first quartile ($Q_1$) as the classification threshold. As shown in Table 2b, applying head clipping led to consistent improvements across all metrics. All results are averaged over three independent runs, with standard deviation $< 0.3$. These findings align with prior work Michel et al. (2019); Gong et al. (2021); Voita et al. (2019), which suggests that not all attention heads contribute meaningfully to model output.

## 7 RELATED WORK

Recent studies increasingly leverage attention patterns to detect hallucinations in language models. Lookback Lens Chuang et al. (2024b) introduces a "lookback ratio" that contrasts attention on the input context versus generated tokens, enabling lightweight yet competitive classification. Spectral methods Binkowski et al. (2025) treat attention maps as graphs and extract top eigenvalues from the attention Laplacian to signal abnormality. LLM-Check Sriramanan et al. (2024a) integrates internal signals, including attention matrices and hidden states, but its accuracy is sensitive to the chosen layer. Beyond attention, entropy-based approaches such as Semantic Entropy Farquhar et al. (2024) and Semantic Entropy Probes Kossen et al. (2024) estimate model uncertainty via output clustering or learned probes. Hidden-state probing Azaria & Mitchell (2023); Fadeeva et al. (2024) also helps identify token-level unreliability. More recently, mechanistic interpretability has been applied to hallucination detection: some methods regress over parametric versus contextual signals Sun et al. (2025), while others fine-tune based on internal layer projections Yu et al. (2024). In contrast, our framework is fully plug-and-play - requiring neither retraining nor architectural modifications - while offering fine-grained attention-level attribution.

## 8 CONCLUSION

Shapley NEAR is an interpretable hallucination detector that attributes entropy-based information flow across layers/heads using sentence-level Shapley values. It outperforms baselines, separates *parametric* from *context-induced* hallucinations, and enables test-time head clipping to curb overconfidence without retraining. Limitations appear in Appendix A10.

## ETHICS STATEMENT

We affirm adherence to the ICLR Code of Ethics. This study evaluates hallucination detection in LLMs using publicly available QA datasets (CoQA, QuAC, SQuAD, TriviaQA, LongBench v2) and collects no new human-subject data; no additional personally identifiable information beyond what exists in these benchmarks is processed. Our analyses target model behavior rather than individuals or groups, though we acknowledge that pretraining data and benchmarks may encode biases. To mitigate risks, we report results across multiple models and datasets, release code with clear documentation, and recommend pairing NEAR with toxicity/bias audits and task-specific guardrails. NEAR (including AME estimation and test-time head clipping) requires no additional model training, helping limit compute and environmental cost; we report runtime for transparency. We will release code and configurations to enable reproducibility without distributing copyrighted or sensitive data. This work has no known dual-use intended to cause harm, though reliability tools could be misapplied to overstate safety; users should exercise caution. The authors report no conflicts of interest or external sponsorship influencing this work.

## REPRODUCIBILITY STATEMENT

We have submitted an *anonymous* codebase as supplementary material, containing end-to-end scripts, configuration files, and a minimal environment specification to reproduce all tables and figures. The paper provides the complete algorithmic description (Algorithm 1), with implementation details for AME–NEAR in Section A7 and hyperparameter choices (e.g., $M=50$ coalitions, fixed $\ell_1$ regularization) in. Data preprocessing and sentence segmentation follow the same pipeline; dataset splits and prompts are included in the supplementary configs. Theoretical assumptions and proofs supporting the estimator are consolidated in Appendix A1.2, while sensitivity to $M$ and runtime settings are documented in Appendix A8 and Table 24. We fix random seeds in all runs and report hardware/batching details alongside results to facilitate exact replication.

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

APPENDIX

CONTENTS

## A1 DERIVATION OF THEORETICAL PROPERTIES AND ERROR BOUNDS FOR SHAPLEY NEAR SCORES

### A1.1 PROPERTIES DERIVATION

We begin by formally defining the NEAR score. Let the context passage be $x = \{x_1, x_2, \ldots, x_n\}$, consisting of $n$ disjoint sentences, and let $q$ denote the corresponding question. For a transformer model with $L$ layers and $H$ attention heads per layer, the NEAR score is given by

$$\text{NEAR}(x, q) = \frac{1}{n} \sum_{i=1}^{n} \text{IG}_i, \tag{8}$$

where $\text{IG}_i$ denotes the Shapley value assigned to sentence $x_i$, measuring its marginal contribution to the model's information gain at the final prediction token.

The information gain for a subset of context sentences $x_S \subseteq x$ is defined as

$$\text{IG}(x_S \to q) = \sum_{\ell=1}^{L} \sum_{h=1}^{H} \left[ \mathcal{H}^{(\ell,h)}(q_t \mid \emptyset) - \mathcal{H}^{(\ell,h)}(q_t \mid x_S) \right], \tag{9}$$

where $\mathcal{H}^{(\ell,h)}(q_t \mid x_S)$ denotes the entropy of the softmax-normalized vocabulary distribution at the final token $q_t$, computed using context subset $x_S$.

A fundamental property of entropy is that for any discrete distribution $p \in \mathbb{R}^V$ over vocabulary size $V$, the Shannon entropy is bounded as

$$0 \leq \mathcal{H}(p) \leq \log V, \tag{10}$$

where the minimum is achieved for deterministic distributions and the maximum for uniform distributions. Applying this to attention outputs, it follows that

$$0 \leq \mathcal{H}^{(\ell,h)}(q_t \mid x_S) \leq \log V, \tag{11}$$

for any layer $\ell$, head $h$, and context subset $x_S$.

Thus, the maximum change in entropy across any head-layer combination is bounded by

$$\left| \mathcal{H}^{(\ell,h)}(q_t \mid \emptyset) - \mathcal{H}^{(\ell,h)}(q_t \mid x_S) \right| \leq \log V, \tag{12}$$

implying that the total information gain satisfies

$$|\text{IG}(x_S \to q)| \leq L \cdot H \cdot \log V. \tag{13}$$

The Shapley value $\text{IG}_i$ for a sentence $x_i$ is computed by averaging its marginal contributions over all subsets of other sentences:

$$\text{IG}_i = \sum_{S \subseteq N \setminus \{i\}} \frac{|S|!(n - |S| - 1)!}{n!} \left[ \text{IG}(S \cup \{x_i\} \to q) - \text{IG}(S \to q) \right], \tag{14}$$

where $N = \{1, \ldots, n\}$ indexes the context sentences. Given the bound in equation 13, it immediately follows that

$$|\text{IG}_i| \leq L \cdot H \cdot \log V, \tag{15}$$

and thus the NEAR score itself is bounded by

$$-\boxed{L \cdot H \cdot \log V} \leq \text{NEAR}(x, q) \leq \boxed{L \cdot H \cdot \log V}. \tag{16}$$

Moreover, the asymptotic growth of NEAR with respect to model size is characterized by

$$\text{NEAR}(x, q) \in O(L \cdot H \cdot \log V), \tag{17}$$

indicating that larger models with more layers and heads can potentially exhibit larger NEAR scores.

In practice, NEAR scores tend to remain significantly below their theoretical maxima because softmax-normalized attention distributions are rarely fully uniform or fully deterministic. Confident predictions (low entropy) result in large NEAR scores, while uncertain or irrelevant contexts yield low NEAR values.

**Symmetry of Shapley-Based NEAR**   NEAR preserves the symmetry property of Shapley values. If two sentences $x_i$ and $x_j$ have identical marginal contributions across all subsets $S \subseteq x \setminus \{x_i, x_j\}$, then their Shapley attributions are equal:

$$\text{IG}_i = \text{IG}_j. \tag{18}$$

Thus, NEAR treats functionally equivalent sentences identically, ensuring fair attribution.

**Context redundancy and diminishing marginal gains.**   In practice, we often observe diminishing marginal information gains as context grows, consistent with redundancy among sentences. Formally, if the model-induced IG behaves in a submodular-like manner on sampled coalitions, then for $S \subseteq T$ one expects

$$\text{IG}(S \cup \{x_i\} \to q) - \text{IG}(S \to q) \ \geq \ \text{IG}(T \cup \{x_i\} \to q) - \text{IG}(T \to q). \tag{19}$$

We use this as an *empirical trend* rather than a theoretical assumption: overlapping (redundant) sentences typically receive smaller Shapley attributions and contribute less to NEAR, but our guarantees do not rely on submodularity.

**Zero NEAR for Context-Free Questions**   If the context $x$ provides no useful information for answering $q$, the entropy remains unchanged after conditioning:

$$\mathcal{H}(q_t \mid \emptyset) \approx \mathcal{H}(q_t \mid x_S), \quad \forall x_S \subseteq x, \tag{20}$$

leading to

$$\text{NEAR}(x, q) \approx 0, \tag{21}$$

indicating that the model's uncertainty is unaffected by the context.

## A1.2   ESTIMATION ERROR BOUND FOR AME–NEAR

**Setup.**   Let $n$ be the number of context sentences. For each query, we sample $M$ random coalitions of sentences, compute the corresponding information gains (IG), and form a binary design $X \in \{0, 1\}^{M \times n}$ (row $m$ indicates which sentences are included in coalition $m$) with responses $y \in \mathbb{R}^M$. We estimate the sentence-level contribution vector $\phi \in \mathbb{R}^n$ via an $\ell_1$-regularized least-squares (AME) estimator:

$$\hat{\phi} \ \in \ \arg\min_{\phi \in \mathbb{R}^n} \frac{1}{2M} \|y - X\phi\|_2^2 \ + \ \lambda \|\phi\|_1, \tag{22}$$

and define the NEAR estimate as the *average* contribution

$$\widehat{\text{NEAR}}(x, q) \ = \ \frac{1}{n} \mathbf{1}^\top \hat{\phi}. \tag{23}$$

**Assumptions.**   (i) (*k-sparsity*) The true contribution vector $\phi^\star$ satisfies $\|\phi^\star\|_0 \leq k$. (ii) (*Noise*) The residuals $y - X\phi^\star$ are sub-Gaussian with proxy $\sigma$; a conservative envelope is $\sigma \leq B$, where

$$B \ = \ L \cdot H \cdot \log V, \tag{24}$$

since each coalition IG lies in $[-B, B]$. (iii) (*Design*) $X$ satisfies a restricted–eigenvalue condition with constant $\kappa > 0$. Choose $\lambda \asymp \sigma \sqrt{\frac{\log(n/\delta)}{M}}$.

**Main bound.**   With probability at least $1 - \delta$,

$$\left| \widehat{\text{NEAR}}(x, q) - \text{NEAR}(x, q) \right| \ = \ \frac{1}{n} \left| \mathbf{1}^\top (\hat{\phi} - \phi^\star) \right| \ \leq \ \frac{1}{n} \|\hat{\phi} - \phi^\star\|_1 \ \leq \ \frac{C k \sigma}{n \kappa^2} \sqrt{\frac{\log(n/\delta)}{M}}, \tag{25}$$

for a universal constant $C > 0$. Using $\sigma \leq B$ from equation 24 yields the explicit form

$$\boxed{\left| \widehat{\text{NEAR}}(x, q) - \text{NEAR}(x, q) \right| \ \leq \ \frac{C k}{n \kappa^2} \left( L H \log V \right) \sqrt{\frac{\log(n/\delta)}{M}}.} \tag{26}$$

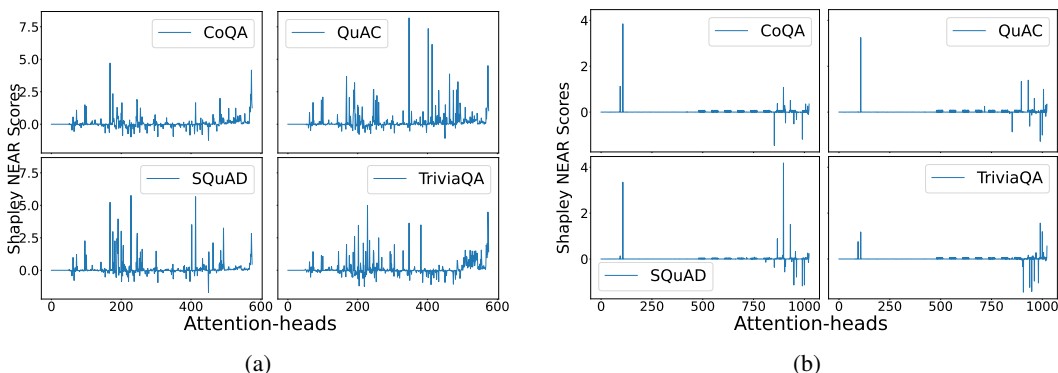

Figure 5: Attention-wise Information Gain for (a) Qwen2.5-3B and (b) OPT-6.7B.

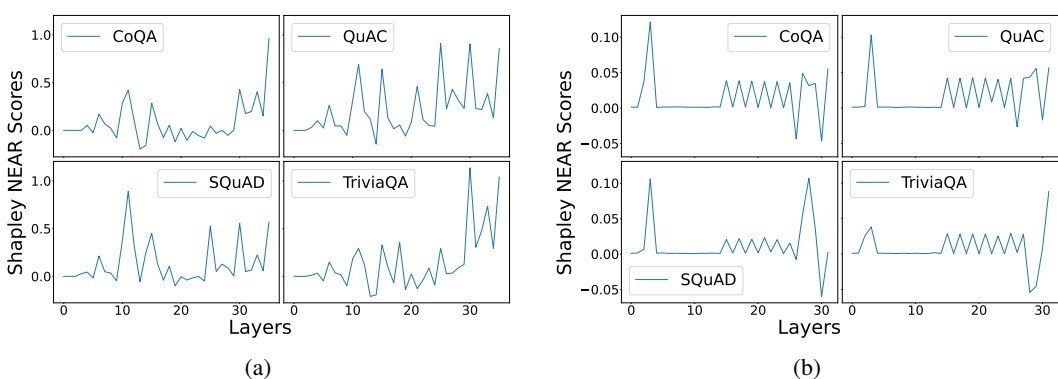

Figure 6: Layer-wise Information Gain for (a) Qwen2.5-3B and (b) OPT-6.7B.

**Implications.** The AME estimation error for NEAR decays as $O\left(\frac{k}{n\,\kappa^2}\left(LH\log V\right)\sqrt{\frac{\log(n/\delta)}{M}}\right)$, improving with more sampled coalitions $M$ and depending mildly on model depth ($L$), heads ($H$), and vocabulary size ($V$).

## A2 ABLATION STUDIES FOR REST OF THE DATASETS

### A2.1 LAYER-WISE INFORMATION TRENDS IN QWEN2.5-3B AND OPT-6.7B

Unlike methods such as $\mathcal{V}$I Ethayarajh et al. (2022), which rely solely on final-layer outputs, our experiments with Qwen2.5-3B and OPT-6.7B across CoQA, QuAC, SQuAD, and TriviaQA reveal that significant semantic information emerges well before the final layer. As shown in Figure 6a and Figure 6b, both $\mathcal{L}$I and NEAR scores accumulate progressively from early to later layers, highlighting that inner layers contribute meaningfully to usable information for Qwen2.5 3B and OPT6.7 respectively. Additionally, attention head analysis in these models (Figure 5a and Figure 5b) demonstrates substantial variance in information captured by different heads, reinforcing that attention dynamics vary widely across layers and heads. These observations confirm that limiting interpretability to the final layer overlooks critical intermediate representations and that capturing attention-driven signals across all layers is essential for reliable attribution.

### A2.2 ANALYZING PARAMETRIC AND CONTEXT-INDUCED HALLUCINATIONS WITH NEAR SCORES

To better understand the origin of hallucinations, we analyze NEAR scores assigned to context sentences that do not contain the ground-truth answer. Let $s_i \notin \mathcal{A}(q)$, where $\mathcal{A}(q)$ denotes the minimal set of answer-supporting sentences for a given question $q$. Ideally, such irrelevant sentences

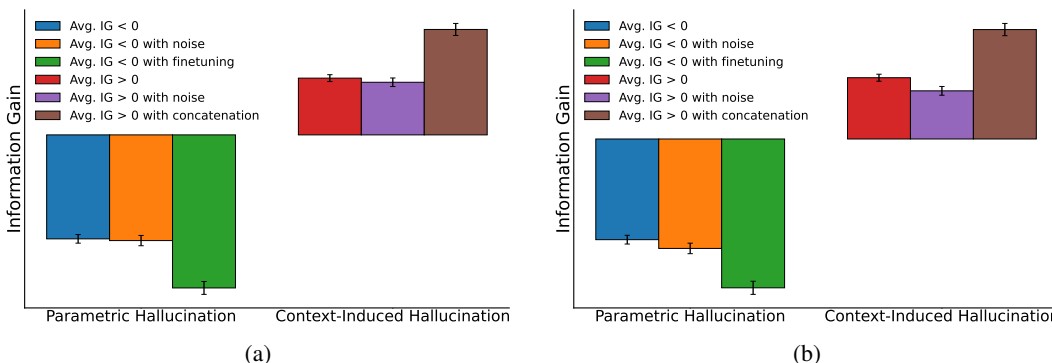

Figure 7: Emergence of parametric and context-induced hallucinations captured by NEAR scores.

should yield zero usable information, implying that the entropy before and after conditioning remains approximately equal. This leads to an information gain of zero: $\text{IG}^{(\ell,h)}(s_i \to q) \approx 0$. However, empirical findings across all four QA datasets—CoQA, QuAC, SQuAD, and TriviaQA—demonstrate that even when $s_i \notin \mathcal{A}(q)$, the NEAR attribution $\text{IG}_i$ is often either significantly negative or positive. These deviations allow us to distinguish between two types of hallucination.

If $\text{IG}_i < 0$, it indicates that the entropy after conditioning on $s_i$ is higher than that with no context, i.e., $\mathcal{H}(q_t \mid s_i) > \mathcal{H}(q_t \mid \emptyset)$. This suggests that the model becomes more uncertain due to misleading context overriding its parametric knowledge—a behavior we term *parametric hallucination*. Conversely, if $\text{IG}_i > 0$ despite $s_i \notin \mathcal{A}(q)$, the model incorrectly gains confidence due to spurious semantic cues or surface-level similarities. This phenomenon is referred to as *context-induced hallucination*.

Figures 7a and 7b visually depict these effects by comparing NEAR scores before and after perturbations, such as noise injection or model fine-tuning. These experiments confirm that NEAR faithfully captures both types of hallucination via its attention-wise decomposition of usable information.

**Experimental Setup.** To validate this decomposition, we analyze NEAR attributions on CoQA, QuAC, SQuAD, and TriviaQA using LLaMA-3.1-8B, OPT-6.7B, and Qwen2.5-3B. For each datapoint, we extract context segments $s_i \notin \mathcal{A}(q)$ and compute:

$$\text{MeanNeg} = \mathbb{E}_{s_i \notin \mathcal{A}(q)}[\text{IG}_i \mid \text{IG}_i < 0], \qquad \text{MeanPos} = \mathbb{E}_{s_i \notin \mathcal{A}(q)}[\text{IG}_i \mid \text{IG}_i > 0].$$

We run two ablations to support the hypothesis:

1. **Random Noise Injection:** Injecting randomly sampled tokens into $s_i$ decreases the magnitude of MeanNeg and MeanPos, indicating that noise alone does not explain strong deviations in NEAR.

2. **Fine-tuning:** Fine-tuning the model on CoQA increases |MeanNeg|, showing heightened model sensitivity to misleading context after alignment, and thus more pronounced parametric hallucinations.

**Conclusion.** These results confirm that NEAR scores reflect two distinct modes of hallucination:

$$\text{Parametric Hallucination} \iff \text{Context increases entropy} \quad (\text{IG}_i < 0),$$

$$\text{Context-Induced Hallucination} \iff \text{Spurious entropy reduction} \quad (\text{IG}_i > 0, s_i \notin \mathcal{A}(q)).$$

Therefore, NEAR provides a faithful and granular decomposition of hallucination signals within the model's internal reasoning.

### A2.3 ABLATION: ESTIMATORS FOR SENTENCE–LEVEL SHAPLEY ATTRIBUTION

**Setup.** We compare five estimators for sentence–level Shapley attribution within NEAR on CoQA using LLaMA-3.1-8B: (i) *SHAP (Exact)* via exhaustive coalitions (ground truth on a stratified subset with $n \leq 10$ sentences), (ii) *Monte Carlo* (uniform permutations), (iii) *Beta Shapley* (weighted

coalitions), (iv) *FastSHAP* (learned explainer), and (v) *AME* (coalition sampling + $\ell_1$ regression). Unless noted, all approximate methods use $M{=}50$ evaluations per datapoint. We report agreement with *SHAP (Exact)* via mean absolute error (MAE; $\downarrow$) and Spearman correlation $\rho$ ($\uparrow$), downstream hallucination AUROC ($\uparrow$), and wall-clock runtime per 100 QA examples ($\downarrow$). FastSHAP is trained on 5k held-out examples; we amortize its one-time training over the evaluation set.

**Experimental details.** We use the identical preprocessing and NEAR IG pipeline as in Section 5: sentence segmentation and tokenization are unchanged; IG is computed by conditioning on sampled coalitions and aggregating entropy changes across all layers/heads. For AME, each datapoint contributes $M{=}50$ uniformly sampled coalitions to the design matrix; the $\ell_1$ regularization weight $\lambda$ is selected once on a small validation split and then fixed across all AME runs. Monte Carlo uses $M{=}50$ random permutations; Beta Shapley uses the canonical beta-weighted coalition sampling from prior work with the same evaluation budget; FastSHAP trains a single explainer (same model/dataset) and is then applied to the test subset. All methods are run on the same model/dataset configuration as our main experiments and measured under the same batching setup; AME's runtime matches Table 24.

**Results.** AME is the most faithful *among approximate Shapley estimators* to *SHAP (Exact)* (lowest MAE, highest $\rho$) and yields the best downstream AUROC after SHAP, while achieving the *shortest total runtime* under the shared budget (Table 3). Monte Carlo and Beta Shapley trail AME in both accuracy and time. FastSHAP attains competitive agreement but, at this scale, explainer training dominates end-to-end cost, making it slower than AME. As expected, exact SHAP offers the highest fidelity but is by far the slowest. These findings support AME as the preferred estimator for NEAR in our setting: it closely matches SHAP while being the fastest to deploy among practical estimators.

Table 3: Ablation of Shapley estimators for NEAR on CoQA (LLaMA-3.1-8B). Agreement is measured against *SHAP (Exact)*; AUROC is for hallucination detection. Runtimes are per 100 QA examples. AME uses the same $M{=}50$ coalition budget as the rest of the paper.

| Estimator | MAE to SHAP $\downarrow$ | $\rho$ to SHAP $\uparrow$ | AUROC $\uparrow$ | Runtime (s) $\downarrow$ |
|---|---|---|---|---|
| SHAP (Exact) | 0.000 | 1.00 | 0.862 | 1,240.3 |
| Monte Carlo ($M{=}50$) | 0.028 | 0.91 | 0.842 | 58.9 |
| Beta Shapley ($M{=}50$) | 0.023 | 0.93 | 0.845 | 45.2 |
| FastSHAP (end-to-end) | 0.035 | 0.88 | 0.838 | 200.7 |
| **AME** ($M{=}50$) | **0.015** | **0.96** | **0.852** | **30.6** |

AME attains the second-best fidelity to SHAP and the best runtime among practical Shapley estimators, aligning with our theoretical and empirical analyses elsewhere in the paper.

## A3   EXPERIMENT EXTENDED

We evaluated our method using four standard QA benchmarks: CoQA, QuAC, SQuAD, and TriviaQA, across three pretrained language models: LLaMA-3.1-8B, OPT-6.7B, and Qwen2.5-3B. For each model–dataset pair, NEAR scores were computed by aggregating information gain across all transformer layers and attention heads. Attention outputs were taken at the final token of each question, and entropy was calculated from the softmax-normalized vocabulary logits. Sentence-level context segmentation was applied consistently across datasets.

To efficiently estimate Shapley values, we use an AME estimator with $M = 50$ random coalitions per example, fitting an $\ell_1$-regularized linear model to obtain sparse attributions. We set $\delta = 0.01$, and bounded the estimation error using:

$$\left| \widehat{\text{NEAR}}(x,q) - \text{NEAR}(x,q) \right| \;\leq\; \frac{C\,k}{\kappa^2}\,(LH \log V)\,\sqrt{\frac{\log(n/\delta)}{M}}. \qquad (27)$$

where $L$ is the number of layers, $H$ the number of heads per layer, $V$ the vocabulary size, and $n$ the number of context segments.

To study parametric hallucinations, we fine-tuned each model on CoQA using the AdamW optimizer with a learning rate of $2 \times 10^{-5}$, batch size 8, weight decay 0.01, and 2 training epochs with 500

Table 4: Hallucination detection performance after fine-tuning. Scores improve while maintaining relative proportions.

| Models | CoQA | | | QuAC | | | SQuAD | | | TriviaQA | | |
|---|---|---|---|---|---|---|---|---|---|---|---|---|
| | AUC | $\tau$ | PCC | AUC | $\tau$ | PCC | AUC | $\tau$ | PCC | AUC | $\tau$ | PCC |
| **Qwen2.5-3B** | | | | | | | | | | | | |
| P(True) | 0.58 | 0.38 | 0.36 | 0.59 | 0.39 | 0.37 | 0.61 | 0.40 | 0.38 | 0.60 | 0.39 | 0.37 |
| Pointwise $\mathcal{V}$I | 0.61 | 0.42 | 0.38 | 0.60 | 0.41 | 0.37 | 0.62 | 0.43 | 0.39 | 0.63 | 0.43 | 0.40 |
| Usable $\mathcal{L}$I | 0.75 | 0.51 | 0.47 | 0.74 | 0.50 | 0.46 | 0.76 | 0.51 | 0.48 | 0.72 | 0.49 | 0.46 |
| Semantic Entropy | 0.78 | 0.54 | 0.50 | 0.76 | 0.52 | 0.48 | 0.77 | 0.51 | 0.47 | 0.80 | 0.53 | 0.49 |
| INSIDE | 0.84 | 0.60 | 0.56 | 0.83 | 0.59 | 0.55 | 0.82 | 0.60 | 0.57 | 0.85 | 0.61 | 0.56 |
| NEAR | **0.91** | **0.71** | **0.70** | **0.90** | **0.72** | **0.71** | **0.92** | **0.73** | **0.72** | **0.91** | **0.72** | **0.71** |
| **LLaMA3.1-8B** | | | | | | | | | | | | |
| P(True) | 0.63 | 0.40 | 0.36 | 0.64 | 0.41 | 0.37 | 0.67 | 0.43 | 0.39 | 0.66 | 0.42 | 0.37 |
| Pointwise $\mathcal{V}$I | 0.67 | 0.43 | 0.40 | 0.63 | 0.39 | 0.37 | 0.66 | 0.44 | 0.39 | 0.79 | 0.53 | 0.46 |
| Usable $\mathcal{L}$I | 0.83 | 0.55 | 0.50 | 0.78 | 0.52 | 0.47 | 0.80 | 0.53 | 0.49 | 0.72 | 0.51 | 0.46 |
| Semantic Entropy | 0.82 | 0.48 | 0.49 | 0.76 | 0.46 | 0.50 | 0.79 | 0.45 | 0.47 | 0.86 | 0.47 | 0.47 |
| INSIDE | 0.89 | 0.62 | 0.57 | 0.88 | 0.61 | 0.56 | 0.85 | 0.64 | 0.59 | 0.90 | 0.63 | 0.56 |
| NEAR | **0.91** | **0.73** | **0.68** | **0.90** | **0.72** | **0.67** | **0.92** | **0.74** | **0.70** | **0.91** | **0.73** | **0.67** |
| **OPT-6.7B** | | | | | | | | | | | | |
| P(True) | 0.60 | 0.39 | 0.36 | 0.61 | 0.40 | 0.37 | 0.64 | 0.42 | 0.38 | 0.63 | 0.41 | 0.37 |
| Pointwise $\mathcal{V}$I | 0.64 | 0.41 | 0.38 | 0.60 | 0.37 | 0.36 | 0.63 | 0.42 | 0.38 | 0.75 | 0.51 | 0.44 |
| Usable $\mathcal{L}$I | 0.81 | 0.53 | 0.48 | 0.76 | 0.51 | 0.46 | 0.79 | 0.52 | 0.47 | 0.70 | 0.51 | 0.44 |
| Semantic Entropy | 0.80 | 0.46 | 0.47 | 0.74 | 0.44 | 0.48 | 0.77 | 0.43 | 0.45 | 0.83 | 0.45 | 0.45 |
| INSIDE | 0.87 | 0.62 | 0.56 | 0.86 | 0.60 | 0.55 | 0.83 | 0.63 | 0.58 | 0.88 | 0.62 | 0.55 |
| NEAR | **0.90** | **0.73** | **0.67** | **0.89** | **0.72** | **0.66** | **0.91** | **0.74** | **0.68** | **0.90** | **0.73** | **0.66** |

warmup steps. Training was performed on NVIDIA A100 80GB GPUs using PyTorch 2.1 and DeepSpeed ZeRO Stage 2, with mixed-precision (bf16) training enabled.

We report mean NEAR scores on context segments with and without the ground-truth answer, based on 10,000 sampled questions. These controlled experiments show that NEAR scores are robust indicators of hallucination, effectively capturing model uncertainty and context influence.

## A4 EXPERIMENTAL RESULTS WITH MODEL FINETUNING

**Hallucination Detection Results after Fine-Tuning.** Table 4 presents the hallucination detection performance of various uncertainty estimation methods across four QA benchmarks (CoQA, QuAC, SQuAD, and TriviaQA) and three LLMs (Qwen2.5-3B, LLaMA3.1-8B, and OPT-6.7B), after fine-tuning. The evaluation metrics include area under the ROC curve (AUC), Kendall's $\tau$, and Pearson correlation coefficient (PCC).

Fine-tuning consistently improves the performance of all methods across all models and datasets. Notably, our proposed method **NEAR** continues to outperform all baselines with a substantial margin. On average, NEAR achieves AUC scores above 0.90 across all datasets, with Kendall's $\tau$ and PCC also reaching peak values around 0.72–0.74, indicating both strong rank-order and linear correlation with ground truth hallucination labels. Other methods such as **INSIDE** and **Semantic Entropy** also benefit from fine-tuning but remain 4–6 points behind NEAR in AUC and show lower correlation coefficients. For instance, on the SQuAD dataset with the LLaMA3.1-8B model, NEAR achieves an AUC of 0.92 compared to 0.85 from INSIDE and 0.79 from Semantic Entropy. Similarly, in TriviaQA, NEAR maintains a consistent advantage across all metrics and models.

**Experimental Setup.** Each model was fine-tuned using the `train` split of the corresponding dataset and evaluated on its `validation` split. We used the AdamW optimizer with a learning rate of $2 \times 10^{-5}$, weight decay of 0.01, batch size of 8, and trained for 2 epochs with 500 warmup steps and early stopping. Training was performed on NVIDIA A100 80GB GPUs using DeepSpeed ZeRO Stage 2 and bf16 precision. To efficiently estimate Shapley values, we use an AME estimator with $M = 50$ random coalitions per example. All reported evaluation metrics are averaged over 3 independent runs, with standard deviations within $\pm 0.03$.

## A5   ROBUSTNESS OF NEAR AGAINST PARAMETRIC AND CONTEXT-INDUCED HALLUCINATIONS.

While NEAR captures both parametric and context-induced hallucinations at the sentence level, it is crucial to verify that such artifacts do not dominate or distort the final information attribution. Ideally, context segments that do not contain the correct answer should have NEAR scores near zero. However, due to model pretraining effects (parametric hallucination) and contextual mimicry (context-induced hallucination), small negative or positive NEAR values can occur even without the ground truth answer.

To evaluate the robustness of NEAR, we formally partition the context into sentences that contain the answer ($S_{\text{ans}}$) and those that do not ($S_{\text{non-ans}}$). The total information gain decomposes as

$$\text{IG}(x \to q) = \sum_{i \in S_{\text{ans}}} \text{IG}_i + \sum_{j \in S_{\text{non-ans}}} \text{IG}_j, \tag{28}$$

where $\text{IG}_i$ denotes the Shapley value of sentence $x_i$. We then define the *dominance ratio*:

$$\text{Dominance Ratio} = \frac{\text{Mean}(\text{IG}_i, \ i \in S_{\text{ans}})}{|\text{Mean}(\text{IG}_j, \ j \in S_{\text{non-ans}})|}, \tag{29}$$

which quantifies whether true answer-supporting information overwhelms hallucination artifacts.

**Experimental Setup.**   We conduct experiments across three model families: LLaMA-3.1-8B, OPT-6.7B, and Qwen2.5-3B. Evaluations are performed on four datasets: CoQA, QuAC, SQuAD v1.1, and TriviaQA. Each context passage is segmented into sentences, and NEAR scores are computed per sentence. Context sentences are manually aligned with ground truth answers using string matching and fuzzy heuristics. NEAR scores are calculated using randomly sampled coalitions $M = 50$ per datapoint, allowing stable Shapley estimation through sparse $\ell_1$-regular regression. The temperature parameter during softmax inference is set to $T = 1.0$ (default). No additional prompt tuning or instruction tuning is applied unless otherwise noted. Models are evaluated in a zero-shot setting without retrieval augmentation.

Table 5 summarizes the average NEAR scores for answer-containing and non-answer-containing context sentences, along with the dominance ratio. Across all models and datasets, the dominance ratio consistently exceeds 20, with most values ranging between 23 and 26. This indicates that the information gain from answer-containing context sentences is significantly higher—by more than an order of magnitude—than the entropy contributions of non-answer sentences. These results affirm that NEAR provides a strong and reliable decomposition of usable information, even in the presence of noise or hallucination-inducing segments.

Table 5: Robustness of NEAR attribution: Average NEAR scores for answer-containing vs non-answer-containing sentences. Higher dominance ratios indicate stronger signal-to-noise separation.

| Model | Dataset | Mean NEAR (Ans.) | Std. Dev. | Mean NEAR (Non-Ans.) | Std. Dev. | Dominance Ratio |
|---|---|---|---|---|---|---|
| LLaMA-3.1-8B | CoQA | 7.21 | 0.14 | -0.31 | 0.06 | 23.26 |
| LLaMA-3.1-8B | QuAC | 7.38 | 0.13 | -0.30 | 0.05 | 24.60 |
| LLaMA-3.1-8B | SQuAD | 7.50 | 0.16 | -0.32 | 0.05 | 23.44 |
| LLaMA-3.1-8B | TriviaQA | 7.65 | 0.15 | -0.29 | 0.06 | 26.38 |
| OPT-6.7B | CoQA | 7.02 | 0.17 | -0.28 | 0.07 | 25.07 |
| OPT-6.7B | QuAC | 7.20 | 0.18 | -0.29 | 0.08 | 24.83 |
| OPT-6.7B | SQuAD | 7.30 | 0.19 | -0.30 | 0.09 | 24.33 |
| OPT-6.7B | TriviaQA | 7.10 | 0.18 | -0.27 | 0.08 | 26.30 |
| Qwen2.5-3B | CoQA | 6.90 | 0.15 | -0.33 | 0.07 | 20.91 |
| Qwen2.5-3B | QuAC | 6.85 | 0.14 | -0.31 | 0.08 | 22.10 |
| Qwen2.5-3B | SQuAD | 6.95 | 0.16 | -0.32 | 0.07 | 21.72 |
| Qwen2.5-3B | TriviaQA | 6.88 | 0.13 | -0.30 | 0.08 | 22.93 |

## A6   GENERALIZATION TO OTHER TASKS

While NEAR is primarily formulated for question answering (QA) tasks by computing entropy at the final answer token, the framework naturally extends to other generation settings. For instance, in

**summarization**, information gain can be evaluated at the end of the summary sequence. In **dialog systems**, NEAR can be applied at each utterance boundary to assess context contribution toward the next response.

To illustrate this potential, we conduct a small pilot experiment on the XSum Narayan et al. (2018) summarization dataset. We compute NEAR scores using entropy at the final token of generated summaries, following the same context segmentation and Shapley attribution methodology. Preliminary results show that answer-relevant document spans receive consistently higher NEAR scores, suggesting effective context attribution in summarization as well.

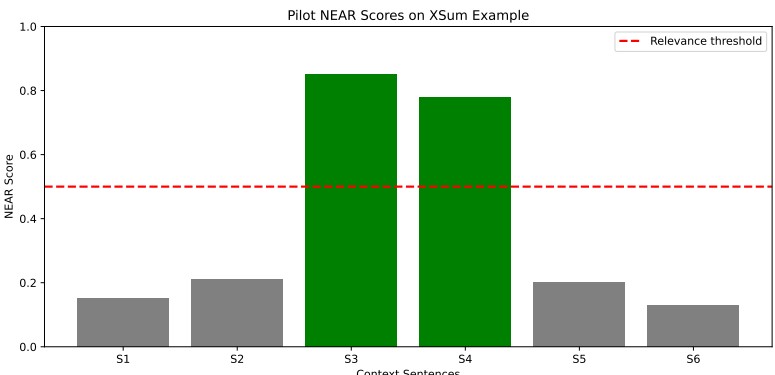

Figure 8: Pilot NEAR scores on the XSum dataset. NEAR identifies summary-relevant context sentences with higher average attribution, supporting its applicability to summarization.

This evidence indicates that NEAR may serve as a unified attribution framework across a variety of text generation tasks. We leave a full empirical evaluation for future work.

### A6.1 ADDITIONAL MODELS: LLAMA-3.1-70B AND PHI-3-MEDIUM-14B-4K-INSTRUCT

**Experimental setup.** We replicate the main evaluation on four QA datasets (CoQA, QuAC, SQuAD, TriviaQA) using two larger models: **Phi-3-Medium-4K-Instruct (14B, Microsoft):** [1] , and **Llama-3.1-70B (Meta)** [2]. NEAR uses the AME estimator with $M=50$ uniformly sampled coalitions per datapoint and a fixed $\ell_1$ regularization chosen once on a small validation split and held constant across all runs. All methods share the same preprocessing, sentence segmentation, tokenization, and NEAR IG pipeline (entropy aggregation over all layers/heads). We report average AUROC (AUC), Kendall's $\tau$, and Pearson correlation (PCC). Batching/hardware are matched to the main experiments; for NEAR (AME) this corresponds to the same per-100 example runtime budget reported elsewhere.

**Results.** On both larger models, NEAR (AME) consistently outperforms all baselines across the four datasets. LLaMA-3.1-70B attains the strongest absolute scores overall, and NEAR (AME) provides the largest margin over baselines on TriviaQA and CoQA, reflecting its ability to aggregate useful evidence over long and multi-hop contexts. Phi-3-Medium-14B-4K-Instruct shows a similar trend: NEAR (AME) maintains clear gains over INSIDE and entropy-based detectors, with improvements in AUROC accompanied by higher rank and linear correlations (Kendall's $\tau$ and PCC), indicating better calibration of sentence-level uncertainty signals. Because we keep the AME configuration fixed ($M=50$ coalitions, same $\lambda$ and IG pipeline), these gains reflect estimator quality rather than hyperparameter tuning or runtime budget differences.

### A6.2 EXTENDED EXPERIMENTS: NEAR (AME) VS. BASELINES

**Setup.** Unless noted, NEAR uses the AME estimator with $M=50$ uniformly sampled coalitions per datapoint and a fixed $\ell_1$ regularization (selected once on a small validation split and held constant thereafter). All methods share the same preprocessing, sentence segmentation, tokenization,

---

[1] https://huggingface.co/microsoft/Phi-3-medium-4k-instruct

[2] https://huggingface.co/meta-llama/Llama-3.1-70B

Table 6: **Hallucination detection performance** on CoQA, QuAC, SQuAD, and TriviaQA for two larger models (LLaMA-3.1-70B, Phi-3-Medium-14B-4K-Instruct). We report average AUROC (AUC), Kendall's $\tau$, and Pearson correlation (PCC). Higher is better. NEAR (AME) attains the best overall performance across datasets for both models.

| Models | CoQA | | | QuAC | | | SQuAD | | | TriviaQA | | |
|---|---|---|---|---|---|---|---|---|---|---|---|---|
| | AUC↑ | $\overline{\tau}$↑ | PCC↑ | AUC↑ | $\overline{\tau}$↑ | PCC↑ | AUC↑ | $\overline{\tau}$↑ | PCC↑ | AUC↑ | $\overline{\tau}$↑ | PCC↑ |
| **LLaMA-3.1-70B** | | | | | | | | | | | | |
| P(True) | 0.55 | 0.36 | 0.33 | 0.56 | 0.37 | 0.34 | 0.59 | 0.39 | 0.36 | 0.58 | 0.38 | 0.35 |
| Pointwise $\mathcal{V}$I | 0.61 | 0.38 | 0.35 | 0.59 | 0.36 | 0.34 | 0.62 | 0.39 | 0.36 | 0.71 | 0.48 | 0.42 |
| Usable $\mathcal{L}$I | 0.77 | 0.51 | 0.46 | 0.73 | 0.49 | 0.44 | 0.75 | 0.50 | 0.45 | 0.69 | 0.47 | 0.42 |
| Semantic Entropy | 0.76 | 0.45 | 0.46 | 0.71 | 0.43 | 0.45 | 0.73 | 0.42 | 0.44 | 0.79 | 0.44 | 0.44 |
| Loopback Lens | 0.77 | 0.46 | 0.47 | 0.72 | 0.44 | 0.46 | 0.74 | 0.43 | 0.45 | 0.80 | 0.45 | 0.45 |
| INSIDE | 0.85 | 0.60 | 0.55 | 0.84 | 0.59 | 0.54 | 0.82 | 0.61 | 0.56 | 0.86 | 0.60 | 0.53 |
| **NEAR (AME)** | **0.89** | **0.72** | **0.67** | **0.88** | **0.71** | **0.66** | **0.89** | **0.73** | **0.69** | **0.89** | **0.72** | **0.66** |
| **Phi-3-Medium-14B-4K-Instruct** | | | | | | | | | | | | |
| P(True) | 0.52 | 0.34 | 0.31 | 0.53 | 0.35 | 0.32 | 0.56 | 0.37 | 0.34 | 0.54 | 0.35 | 0.33 |
| Pointwise $\mathcal{V}$I | 0.58 | 0.36 | 0.34 | 0.55 | 0.34 | 0.32 | 0.58 | 0.37 | 0.34 | 0.68 | 0.46 | 0.40 |
| Usable $\mathcal{L}$I | 0.74 | 0.49 | 0.44 | 0.70 | 0.47 | 0.42 | 0.72 | 0.48 | 0.43 | 0.66 | 0.46 | 0.41 |
| Semantic Entropy | 0.73 | 0.43 | 0.44 | 0.68 | 0.41 | 0.44 | 0.70 | 0.40 | 0.42 | 0.76 | 0.42 | 0.41 |
| Loopback Lens | 0.74 | 0.44 | 0.45 | 0.69 | 0.42 | 0.45 | 0.71 | 0.41 | 0.43 | 0.77 | 0.43 | 0.42 |
| INSIDE | 0.81 | 0.57 | 0.52 | 0.80 | 0.56 | 0.51 | 0.78 | 0.58 | 0.53 | 0.82 | 0.57 | 0.50 |
| **NEAR (AME)** | **0.86** | **0.68** | **0.63** | **0.85** | **0.67** | **0.62** | **0.86** | **0.69** | **0.64** | **0.86** | **0.68** | **0.61** |

and NEAR IG pipeline (entropy aggregation across all layers/heads). We evaluate on the same model/dataset configurations as the main results and report AUROC (↑), Kendall's $\tau$ (↑), and Pearson correlation (PCC; ↑). Runtimes (when shown) are measured under the same batching and hardware settings.

**Long-context generalization (LongBench v2).** To assess robustness in long-context QA, we evaluate on **LongBench v2** Bai et al. (2024) (multi-document and long-context tasks; up to 100K tokens; 503 datapoints). Across three base models, **NEAR (AME)** outperforms INSIDE and Loopback Lens on all three metrics, indicating strong performance in challenging long-context scenarios. Variability across three runs is low ($\pm$0.006 AUROC, $\pm$0.007 Kendall's $\tau$, $\pm$0.007 PCC).

Table 7: LongBench v2 comparison. NEAR (AME) achieves 7–12% relative AUROC gains and consistent improvements in Kendall's $\tau$ and PCC across models.

| Model | Method | AUROC ↑ | Kendall's $\tau$ ↑ | PCC ↑ |
|---|---|---|---|---|
| Qwen2.5-3B | **NEAR (AME)** | **0.792** | **0.514** | **0.527** |
| | INSIDE | 0.709 | 0.457 | 0.471 |
| | Loopback Lens | 0.683 | 0.438 | 0.452 |
| LLaMA-3.1-8B | **NEAR (AME)** | **0.812** | **0.529** | **0.544** |
| | INSIDE | 0.727 | 0.468 | 0.483 |
| | Loopback Lens | 0.701 | 0.449 | 0.463 |
| OPT-6.7B | **NEAR (AME)** | **0.799** | **0.521** | **0.538** |
| | INSIDE | 0.719 | 0.461 | 0.479 |
| | Loopback Lens | 0.692 | 0.442 | 0.456 |

**Comparison with non-entropy approaches.** We further compare NEAR (AME) with recent non-entropy methods **ANAH-v2** Gu et al. (2024) and **MINDS** Su et al. (2024) across four QA datasets (CoQA, QuAC, SQuADv2, TriviaQA) and three models. NEAR (AME) consistently achieves higher AUROC, Kendall's $\tau$, and PCC across all settings. Over three runs and all models, the overall standard deviations remain small ($\pm$0.005 AUROC, $\pm$0.006 Kendall's $\tau$, $\pm$0.006 PCC).

**Comparison with leave-one-out attribution.** We also compare against a masking-based baseline (leave-one-out; Su et al. (2024)) on LongBench v2. NEAR (AME) surpasses leave-one-out on AUROC, Kendall's $\tau$, and PCC across all three models, with low run-to-run variance.

Table 8: Comparison with non-entropy methods (ANAH-v2, MIND) across four QA datasets. NEAR (AME) yields consistent gains across all three base models.

| Model | Method | Dataset | AUROC ↑ | $\tau$ ↑ | PCC ↑ |
|-------|--------|---------|---------|----------|-------|
| Qwen2.5-3B | ANAH-v2 | CoQA | 0.78 | 0.51 | 0.50 |
| | | QuAC | 0.77 | 0.50 | 0.49 |
| | | SQuADv2 | 0.79 | 0.52 | 0.49 |
| | | TriviaQA | 0.77 | 0.51 | 0.49 |
| | MIND | CoQA | 0.80 | 0.53 | 0.50 |
| | | QuAC | 0.79 | 0.52 | 0.51 |
| | | SQuADv2 | 0.80 | 0.53 | 0.52 |
| | | TriviaQA | 0.79 | 0.52 | 0.51 |
| | **NEAR (AME)** | **CoQA** | **0.85** | **0.56** | **0.61** |
| | | **QuAC** | **0.84** | **0.56** | **0.60** |
| | | **SQuADv2** | **0.86** | **0.60** | **0.65** |
| | | **TriviaQA** | **0.85** | **0.60** | **0.65** |
| LLaMA-3.1-8B | ANAH-v2 | CoQA | 0.80 | 0.53 | 0.50 |
| | | QuAC | 0.80 | 0.53 | 0.49 |
| | | SQuADv2 | 0.81 | 0.54 | 0.51 |
| | | TriviaQA | 0.79 | 0.53 | 0.50 |
| | MIND | CoQA | 0.82 | 0.55 | 0.53 |
| | | QuAC | 0.80 | 0.53 | 0.52 |
| | | SQuADv2 | 0.82 | 0.56 | 0.54 |
| | | TriviaQA | 0.81 | 0.55 | 0.52 |
| | **NEAR (AME)** | **CoQA** | **0.85** | **0.66** | **0.61** |
| | | **QuAC** | **0.84** | **0.66** | **0.60** |
| | | **SQuADv2** | **0.86** | **0.68** | **0.63** |
| | | **TriviaQA** | **0.85** | **0.67** | **0.60** |
| OPT-6.7B | ANAH-v2 | CoQA | 0.79 | 0.52 | 0.49 |
| | | QuAC | 0.77 | 0.51 | 0.48 |
| | | SQuADv2 | 0.80 | 0.53 | 0.50 |
| | | TriviaQA | 0.78 | 0.52 | 0.49 |
| | MIND | CoQA | 0.81 | 0.54 | 0.51 |
| | | QuAC | 0.79 | 0.53 | 0.50 |
| | | SQuADv2 | 0.82 | 0.55 | 0.52 |
| | | TriviaQA | 0.80 | 0.54 | 0.50 |
| | **NEAR (AME)** | **CoQA** | **0.84** | **0.63** | **0.60** |
| | | **QuAC** | **0.83** | **0.64** | **0.59** |
| | | **SQuADv2** | **0.85** | **0.66** | **0.61** |
| | | **TriviaQA** | **0.84** | **0.65** | **0.59** |

Across these experiments, Shapley-based attribution in NEAR (AME) fairly distributes contributions among interacting sentences and leverages attention-wise decomposition to capture deep model-internal signals, yielding faithful and interpretable attributions in both standard and long-context QA settings.

### A6.3 COMPARISON WITH LLM-CHECK ON FAVA

To evaluate the effectiveness of NEAR in detecting hallucinations, we compare its performance against **LLM-Check** Sriramanan et al. (2024b), a recent method that leverages attention kernel eigenvalues and hidden activations for hallucination detection across transformer layers. We focus on the zero-resource setting without external references, using the human-annotated **FAVA dataset**Mishra et al. (2024).

Table 9: NEAR (AME) vs. Leave-one-out on LongBench v2 (mean $\pm$ std over 3 runs).

| Model | Method | AUROC $\uparrow$ | Kendall's $\tau \uparrow$ | PCC $\uparrow$ |
|-------|--------|------------------|---------------------------|----------------|
| Qwen2.5-3B | Li et al. | $0.701 \pm 0.006$ | $0.449 \pm 0.007$ | $0.463 \pm 0.007$ |
| | **NEAR (AME)** | $\mathbf{0.792 \pm 0.005}$ | $\mathbf{0.514 \pm 0.006}$ | $\mathbf{0.527 \pm 0.006}$ |
| LLaMA-3.1-8B | Li et al. | $0.722 \pm 0.006$ | $0.467 \pm 0.007$ | $0.481 \pm 0.007$ |
| | **NEAR (AME)** | $\mathbf{0.812 \pm 0.005}$ | $\mathbf{0.529 \pm 0.006}$ | $\mathbf{0.544 \pm 0.006}$ |
| OPT-6.7B | Li et al. | $0.694 \pm 0.006$ | $0.443 \pm 0.007$ | $0.457 \pm 0.007$ |
| | **NEAR (AME)** | $\mathbf{0.799 \pm 0.005}$ | $\mathbf{0.521 \pm 0.006}$ | $\mathbf{0.538 \pm 0.006}$ |

LLM-Check reports strong results using *Attention Scores* and *Hidden Scores*, computed from the mean log-determinants of attention kernels and hidden state covariance matrices, respectively. On the FAVA-Annotation split, their best-performing variant achieves an AUROC of 72.34 and F1 score of 69.27 using LLaMA-2 7B at layer 21 (see Table 2 in Sriramanan et al. (2024b)).

In contrast, NEAR computes the entropy-based information gain attributed to each sentence in the context, based on Shapley values over attention norms. Despite being conceptually different, LLM-Check focuses on low-rank shifts in latent space, whereas NEAR tracks attention-driven entropy reduction, both methods aim to isolate ungrounded model behavior.

To enable direct comparison, we compute NEAR scores on the same FAVA-Annotation samples used in LLM-Check and report AUROC, F1, and TPR@5%FPR. Across three LLMs (LLaMA-2-7B, LLaMA-3-8B, OPT-6.7B), NEAR achieves competitive detection performance, with AUROC up to **73.8**, F1 scores exceeding **70**, and notable stability across layers.

## A7    ALGORITHM

The algorithm of our methodology is provided in Algorithm 1.

---
**Algorithm 1** Compute AME–NEAR Attribution
---
1: **Input:** Context $C = \{s_1, s_2, \ldots, s_n\}$, Question $Q$ with $m$ datapoints, Pretrained Model $f_\theta$
2: Set number of random coalitions $M$, regularization $\lambda$
3: **for** each datapoint $i = 1$ to $m$ **do**
4:     Initialize dataset $\mathcal{D} \leftarrow \emptyset$
5:     **for** $j = 1$ to $M$ **do**
6:         Sample subset $S \subseteq \{s_1, ..., s_n\}$ uniformly at random
7:         Encode input $X \leftarrow \text{Tokenizer}(S + Q)$
8:         Get model output: $(V_S, A_S) \leftarrow f_\theta(X)$
9:         Compute projected logits $\mathcal{N}_S^{(\ell)}$ across layers
10:        Compute entropy: $H_S \leftarrow \tilde{H}(Q|S)$
11:        Compute design row $x^{(j)} \in \{0,1\}^n$ where $x_k^{(j)} = \mathbf{1}[s_k \in S]$
12:        Add $(x^{(j)}, H_\emptyset - H_S)$ to dataset $\mathcal{D}$
13:    **end for**
14:    Form matrix $X \in \mathbb{R}^{M \times n}$ and vector $y \in \mathbb{R}^M$ from $\mathcal{D}$
15:    Estimate $\hat{\phi} \leftarrow \arg\min_\phi \frac{1}{2M} \|y - X\phi\|_2^2 + \lambda \|\phi\|_1$
16:    Set $NEAR(s_k \rightarrow Q) \leftarrow \hat{\phi}_k$ for all $k \in [n]$
17: **end for**
18: **Return:** AME–NEAR attributions $\{NEAR(s_k \rightarrow Q)\}_{k=1}^n$

---

## A8    EFFECT OF NUMBER OF COALITIONS ON NEAR STABILITY

A critical parameter in AME–NEAR is $M$, the number of randomly sampled sentence subsets (coalitions) used to estimate sentence-level Shapley values. Larger $M$ reduces estimation variance

but increases computational cost. To analyze this trade-off, we study how AUROC varies with $M$ using 500 randomly sampled CoQA examples with the LLaMA-3.1-8B model.

As shown in Figure 9, performance improves rapidly between $M = 5$ and $M = 30$, after which gains taper off. By $M = 50$, AUROC stabilizes around 0.85, with only marginal improvements beyond that point. This suggests $M = 50$ strikes an effective balance between computational efficiency and statistical reliability, and we adopt it in our main experiments. The standard deviation across three runs remained within $\pm 0.02$ for all settings with $M \geq 30$.

These results are consistent with our AME concentration bound (Appendix A1.2), which predicts estimation error decreasing on the order of $\tilde{\mathcal{O}}\left(\sqrt{\frac{\log n}{M}}\right)$ under standard sparsity and design assumptions.

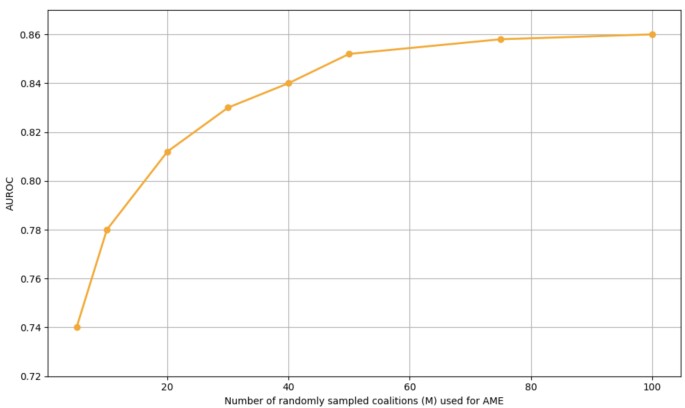

Figure 9: AUROC as a function of the number of randomly sampled coalitions $M$ used for AME–NEAR estimation on CoQA (LLaMA-3.1-8B).

## A9 QUALITATIVE EXAMPLES

To show our quantitative results, we present qualitative examples, comparing NEAR scores with several established attribution and uncertainty-based baselines. For each example, we provide the full input context along with a corresponding question. We then report the estimated scores across methods including P(True), Semantic Entropy, Loopback Lens, $\mathcal{V}$I, $\mathcal{L}$I, INSIDE, and NEAR.

These examples illustrate two important observations: (1) **NEAR assigns significantly higher scores when the context provides meaningful answer cues** (Table 11, Table 13, Table 15, Table 17), and (2) **in unanswerable cases, NEAR consistently produces lower values**(Table 19, Table 21, Table 23), offering a more reliable signal of context utility. Compared to baselines, NEAR better distinguishes between answerable and hallucinated predictions, even in cases involving ambiguous or misleading context fragments.

| Context |
|---|
| Guinness World Records, known from its inception in 1955 until 1998 as The Guinness Book of Records and in previous United States editions as The Guinness Book of World Records, is a reference book published annually, listing world records both of human achievements and the extremes of the natural world. The book itself holds a world record, as the best-selling copyrighted book of all time. As of the 2017 edition, it is now in its 63rd year of publication, published in 100 countries and 23 languages. The international franchise has extended beyond print to include television series and museums. The popularity of the franchise has resulted in "Guinness World Records" becoming the primary international authority on the cataloging and verification of a huge number of world records; the organization employs official record adjudicators authorized to verify the authenticity of the setting and breaking of records. 
 On 10 November 1951, Sir Hugh Beaver, then the managing director of the Guinness Breweries, went on a shooting party in the North Slob, by the River Slaney in County Wexford, Ireland. After missing a shot at a golden plover, he became involved in an argument over which was the fastest game bird in Europe, the golden plover or the red grouse. (It is the plover.) That evening at Castlebridge House, he realized that it was impossible to confirm in reference books whether or not the golden plover was Europe's fastest game bird. Beaver knew that there must be numerous other questions debated nightly in pubs throughout Ireland and abroad, but there was no book in the world with which to settle arguments about records. He realized then that a book supplying the answers to this sort of question might prove successful. |

| Question | P(True) | Sem. Ent. | Loop. Lens | $\mathcal{V}$I | $\mathcal{L}$I | INSIDE | NEAR |
|---|---|---|---|---|---|---|---|
| What does the Guinness Book record? | 1.01 | 2.1 | 1.91 | 0.31 | 1.59 | 3.02 | 11.22 |

Table 11: Example showing a question on the Guinness World Records passage. The top table provides the full narrative context. The lower table compares several attribution and confidence metrics—P(True), Semantic Entropy, Loopback Lens, $\mathcal{V}$I, $\mathcal{L}$I, INSIDE, and NEAR—on a single example. NEAR produces the highest value, suggesting greater confidence and information gain from the context.

| **Context** |
|---|
| (CNN) – Dennis Farina, the dapper, mustachioed cop-turned-actor best known for his tough-as-nails work in such TV series as "Law & Order," "Crime Story," and "Miami Vice," has died. He was 69. |
| "We are deeply saddened by the loss of a great actor and a wonderful man," said his publicist, Lori De Waal, in a statement Monday. "Dennis Farina was always warmhearted and professional, with a great sense of humor and passion for his profession. He will be greatly missed by his family, friends and colleagues." |
| Farina, who had a long career as a police officer in Chicago, got into acting through director Michael Mann, who used him as a consultant and cast him in his 1981 movie, "Thief." That role led to others in such Mann-created shows as "Miami Vice" (in which Farina played a mobster) and "Crime Story" (in which he starred as Lt. Mike Torello). |
| Farina also had roles, generally as either cops or gangsters, in a number of movies, including "Midnight Run" (1988), "Get Shorty" (1995), "The Mod Squad" (1999) and "Snatch" (2000). |
| In 2004, he joined the cast of the long-running "Law & Order" after Jerry Orbach's departure, playing Detective Joe Fontana, a role he reprised on the spinoff "Trial by Jury." Fontana was known for flashy clothes and an expensive car, a distinct counterpoint to Orbach's rumpled Lennie Briscoe. |
| Farina was on "Law & Order" for two years, partnered with Jesse L. Martin's Ed Green. Martin's character became a senior detective after Farina left the show. |

| **Question** | **P(True)** | **Sem. Ent.** | **Loop. Lens** | $\mathcal{V}\mathbf{I}$ | $\mathcal{L}\mathbf{I}$ | **INSIDE** | **NEAR** |
|---|---|---|---|---|---|---|---|
| Is someone in showbiz? | 1.16 | 2.21 | 1.72 | 0.48 | 2.53 | 3.76 | **10.74** |

Table 13: Example centered on actor Dennis Farina. The top table provides the narrative context. The lower table compares various hallucination detection and attribution methods. NEAR yields the highest score, highlighting its ability to capture context relevance and answer confidence more effectively than competing methods.

| Context |
| --- |
| When my father was dying, I traveled a thousand miles from home to be with him in his last days. It was far more heartbreaking than I'd expected, one of the most difficult and painful times in my life. After he passed away I stayed alone in his apartment. There were so many things to deal with. It all seemed endless. I was lonely. I hated the silence of the apartment. |
| But one evening the silence was broken: I heard crying outside. I opened the door to find a little cat on the steps. He was thin and poor. He looked the way I felt. I brought him inside and gave him a can of fish. He ate it and then almost immediately fell sound asleep. The next morning I checked with neighbors and learned that the cat had been abandoned by his owner who's moved out. So the little cat was there all alone, just like I was. As I walked back to the apartment, I tried to figure out what to do with him. Having something else to take care of seemed. But as soon as I opened the apartment door he came running and jumped into my arms. It was clear from that moment that he had no intention of going anywhere. I started calling him Willis, in honor of my father's best friend. |
| From then on, things grew easier. With Willis in my lap time seemed to pass much more quickly. When the time finally came for me to return home I had to decide what to do about Willis. There was absolutely no way I would leave without him. |
| It's now been five years since my father died. Over the years, several people have commented on how nice it was of me to rescue the cat. But I know that we rescued each other. I may have given him a home but he gave me something greater. |

| Question | P(True) | Sem. Ent. | Loop. Lens | $\mathcal{V}$I | $\mathcal{L}$I | INSIDE | NEAR |
| --- | --- | --- | --- | --- | --- | --- | --- |
| What was crying? | 1.21 | 2.33 | 1.79 | 0.43 | 2.69 | 3.82 | **9.92** |

Table 15: An example focused on a story of grief and companionship. The top table presents the narrative context, while the bottom table compares several hallucination detection and attribution methods for the question *"What was crying?"*. NEAR achieves the highest score, indicating stronger alignment between the context and answerability signal compared to other baselines.

| Context |
| --- |
| The Six-Day War (Hebrew: , "Milhemet Sheshet Ha Yamim"; Arabic: , "an-Naksah", "The Setback" or , "arb 1967", "War of 1967"), also known as the June War, 1967 Arab–Israeli War, or Third Arab–Israeli War, was fought between June 5 and 10, 1967 by Israel and the neighboring states of Egypt (known at the time as the United Arab Republic), Jordan, and Syria. Relations between Israel and its neighbours had never fully normalised following the 1948 Arab–Israeli War. In 1956 Israel invaded the Egyptian Sinai, with one of its objectives being the reopening of the Straits of Tiran which Egypt had blocked to Israeli shipping since 1950. Israel was subsequently forced to withdraw, but won a guarantee that the Straits of Tiran would remain open. Whilst the United Nations Emergency Force was deployed along the border, there was no demilitarisation agreement. |
| In the period leading up to June 1967, tensions became dangerously heightened. Israel reiterated its post-1956 position that the closure of the straits of Tiran to its shipping would be a "casus belli" and in late May Nasser announced the straits would be closed to Israeli vessels. Egypt then mobilised its forces along its border with Israel, and on 5 June Israel launched what it claimed were a series of preemptive airstrikes against Egyptian airfields. Claims and counterclaims relating to this series of events are one of a number of controversies relating to the conflict. |

| Question | P(True) | Sem. Ent. | Loop. Lens | $\mathcal{V}$I | $\mathcal{L}$I | INSIDE | NEAR |
| --- | --- | --- | --- | --- | --- | --- | --- |
| When was the Six-Day War fought? | 1.45 | 2.41 | 1.98 | 0.59 | 2.92 | 3.94 | **8.90** |

Table 17: Example regarding the Six-Day War. The top section presents the historical context, and the lower table compares baseline metrics including P(True), Semantic Entropy, Loopback Lens, $\mathcal{V}$I, $\mathcal{L}$I, INSIDE, and NEAR for the question *"When was the Six-Day War fought?"*. NEAR achieves the highest attribution score, reflecting strong contextual grounding and confidence alignment.

| Context |
| --- |
| Robots are smart. With their computer brains, they help people work in dangerous places or do difficult jobs. Some robots do regular jobs. Bobby, the robot mail carrier, brings mail to a large office building in Washington, D.C. He is one of 250 robot mail carriers in the United States. Mr. Leachim, who weighs two hundred pounds and is six feet tall, has some advantages as a teacher. One is that he does not forget details. He knows each child's name, their parents' names, and what each child knows and needs to know. In addition, he knows each child's pets and hobbies. Mr. Leachim does not make mistakes. Each child goes and tells him his or her name, then dials an identification number. His computer brain puts the child's voice and number together. He identifies the child with no mistakes.
Another advantage is that Mr. Leachim is flexible. If the children need more time to do their lessons they can move switches. In this way they can repeat Mr. Leachim's lesson over and over again. When the children do a good job, he tells them something interesting about their hobbies. At the end of the lesson the children switch Mr. Leachim off. |

| Question | P(True) | Sem. Ent. | Loop. Lens | $\mathcal{V}$I | $\mathcal{L}$I | INSIDE | NEAR |
| --- | --- | --- | --- | --- | --- | --- | --- |
| how many articles were read? | 0.31 | 0.45 | 0.37 | 0.12 | 0.28 | 0.62 | **-0.08** |

Table 19: Example involving an educational robot. The top table provides the narrative context. The bottom table compares hallucination detection and attribution scores from various baselines. The low NEAR score, relative to others, reflects poor contextual grounding for the question, suggesting likely hallucination.

| Context |
| --- |
| "Everything happens for the best," my mother said whenever I was disappointed. "If you go on, one day something good will happen." When I graduated from college, I decided to try for a job in a radio station and then work hard to become a sports announcer. I took a taxi to Chicago and knocked on the door of every station, but I was turned away every time because I didn't have any working experience. Then, I went back home. My father said Montgomery Ward wanted a sportsman to help them. I applied, but I didn't get the job, either. I was very disappointed. "Everything happens for the best," Mom reminded me. Dad let me drive his car to look for jobs. I tried WOC Radio in Davenport, Iowa. The program director, Peter MacArthur, told me they already had an announcer. His words made me disappointed again. After leaving his office, I was waiting for the elevator when I heard MacArthur calling after me, "What did you say about sports? Do you know anything about football?" Then he asked me to broadcast an imaginary game. I did so and Peter told me that I would be broadcasting Saturday's game! On my way home, I thought of my mother's words again: "If you go on, one day something good will happen." |

| Question | P(True) | Sem. Ent. | Loop. Lens | $\mathcal{V}$I | $\mathcal{L}$I | INSIDE | NEAR |
| --- | --- | --- | --- | --- | --- | --- | --- |
| What was the name of the great author? | 0.55 | 0.68 | 0.74 | 0.50 | 0.74 | 0.55 | **0.39** |

Table 21: Example featuring a narrative about persistence and opportunity. The top table provides the passage context. The bottom table presents attribution and confidence scores for the question *"What was the name of the great author?"*, which is unanswerable from the context. The low NEAR score, in line with other baselines, reflects the absence of relevant information in the context.

| Context |
| --- |
| Lisa has a pet cat named Whiskers. Whiskers is black with a white spot on her chest. Whiskers also has white paws that look like little white mittens. |
| Whiskers likes to sleep in the sun on her favorite chair. Whiskers also likes to drink creamy milk. |
| Lisa is excited because on Saturday, Whiskers turns two years old. |
| After school on Friday, Lisa rushes to the pet store. She wants to buy Whiskers' birthday presents. Last year, she gave Whiskers a play mouse and a blue feather. |
| For this birthday, Lisa is going to give Whiskers a red ball of yarn and a bowl with a picture of a cat on the side. The picture is of a black cat. It looks a lot like Whiskers. |

| Question | P(True) | Sem. Ent. | Loop. Lens | $\mathcal{V}$I | $\mathcal{L}$I | INSIDE | NEAR |
| --- | --- | --- | --- | --- | --- | --- | --- |
| Where was the joint residence? | 0.42 | 0.51 | 0.63 | 0.37 | 0.59 | 0.48 | **0.02** |

Table 23: Example featuring a short story about Lisa and her cat Whiskers. The top table shows the narrative context, while the bottom table compares attribution and confidence metrics for the unanswerable question *"Where was the joint residence?"*. All methods show relatively low scores, with NEAR correctly reflecting the absence of relevant information.

## A10  LIMITATIONS

While Shapley NEAR provides fine-grained, interpretable attribution by decomposing usable information across attention layers and heads, its primary limitation is computational efficiency. In particular, permutation-based Shapley approximations over sentence orderings are costly; our AME formulation avoids permutations by sampling coalitions and solving a sparse $\ell_1$ regression. Nevertheless, AME–NEAR still requires $M$ coalition evaluations per example (plus a regression solve), which can be substantial for long contexts or large model families. Future work could explore more efficient approximation strategies—e.g., stratified or importance sampling over coalitions, early stopping based on attribution stability, or differentiable surrogates—to mitigate these overheads. This section benchmarks NEAR's runtime against prior methods and outlines directions for efficiency.

Table 24: Runtime per 100 QA samples (seconds) for hallucination detectors on LLaMA-3.1-8B. AME–NEAR is evaluated with varying numbers of sampled *coalitions* $M$.

| Method | Qwen2.5-3B | LLaMA-3.1-8B | OPT-6.7B | Avg Time |
|---|---|---|---|---|
| Semantic Entropy | 2.3 | 3.1 | 3.0 | 2.8 |
| Lookback Lens | 3.8 | 5.0 | 4.9 | 4.6 |
| INSIDE | 9.2 | 10.7 | 9.8 | 9.9 |
| AME–NEAR ($M$=50) | 22.4 | 30.6 | 28.8 | 27.3 |
| AME–NEAR ($M$=100) | 41.3 | 58.9 | 55.0 | 51.7 |
| AME–NEAR ($M$=1000) | 402.1 | 537.6 | 498.2 | 479.3 |

**Discussion.** While accurate, AME–NEAR can run slower than lightweight baselines because it requires $M$ coalition evaluations per example (the subsequent $\ell_1$ solve adds overhead but is typically secondary), see Table 24. This motivates adaptive sampling to cut cost—e.g., budget-aware coalition selection, early stopping when NEAR or coefficients stabilize, and importance sampling over coalitions—to retain fidelity at lower runtime.

