# OpenReview forum: "Fact or Hallucination? An Entropy-Based Framework for Attention-Wise Usable Information in LLMs"
_ICLR.cc/2026/Conference — ICLR 2026 Conference Withdrawn Submission_

### Official Review · Reviewer_m92T · 2025-10-16

**Soundness:** 3
**Presentation:** 2
**Contribution:** 3
**Rating:** 6
**Confidence:** 3

**Summary:**

The authors propose a new method for hallucination detection of LLM outputs.
The method uses the usable information provided from one layer to the next and also allows for interpretability by distinguishing between hallucinations stemming from parametric knowledge or inputs.
The method outperforms relevant baselines on a variety of datasets.

**Strengths:**

- The results are quite impressive with clear improvements over recent and relevant baselines on a variety of benchmarks.

- The authors also did a good job ablating their design choices and giving intuitions into the behavior of the method.

- I find the method itself quite well-motivated and intuitive to understand.

**Weaknesses:**

- The notation and writing are at times a bit confusing. At times it was hard to follow what variables (e.g. q, s, x), also in combination with subscripts, all stand for and some quantities like p where overloaded often which made it a bit hard to follow. Perhaps the writing of Sec. 3 could be made a bit more clearly and intuitive. There is also caligraphic V for model (which is often used for vocabulary in NLP) and V for vocabulary which is a bit confusing, though admittedly the authors inherited this confusion from (caligraphic) V-information.

- The presentation could also be improved a bit, some figures like Fig. 3 are quite small and hard to read.

- The method is a bit slow compared to baselines (but can still be run in realtime, faster than 1s per query) which is a bit hidden in the appendix but could be moved to the main paper for transparency.

**Questions:**

- The overloading of p around Eq. 3 is a bit confusing, could you maybe explain p_i a bit more clearly, are q and x perhaps missing on the RHS?

- L247: "Equation equation" -> perhaps you are mixing the usage of \ref and \cref? In the paragraph from 265-269 there is only the number as a hyperlink.

- If s_i is a sentence and the score of Eq. 7 is summed over the i does it make sense to call it "sentence-level"? Shouldn't it be passage-level?

- L297+298: is it correct that the inequality is not proven rigorously by the authors but an empirical "law"? If not you might want to add a proof and make it clear that it indeed holds in general (or in a special case).

---

> ### Author Response · Authors · 2025-11-23
>
> Weakness 1) Thank you for this comment. All core variables are formally defined in Section 2 (background) and Section 3 (Shapley NEAR), and we use the same notation consistently in the theoretical appendix, but we agree that the density of symbols (x, q, sᵢ, sₓ) and the overloading of p can make Section 3 harder to read on a first pass. Likewise, using calligraphic 𝒱 for the model family and V for the vocabulary, while standard in the V-information literature, can be confusing in an NLP setting. In the revision, we will (i) add a concise notation table at the beginning of Section 3, (ii) rename the model family (e.g., from 𝒱 to 𝓜) and reserve V exclusively for the vocabulary, and (iii) reduce overloading of p and simplify subscripts to make the exposition more intuitive. These changes will clarify the notation without altering any of the underlying definitions or results.
>
> Weakness 2) Noted with thanks
>
> Weakness 3) Noted with thanks. We didnt hid it but forced to put in Appendix for space constraint by linking it in the main paper.
>
> Thank you for these detailed suggestions. We address them pointwise below.
>
> Question1) Noted with tanks. In our intent, pᵢ denotes the i-th component of the softmax distribution over the vocabulary produced at layer/head (ℓ, h) for the final token, i.e.,
>    pᵢ = p^{(ℓ,h)}(vᵢ | q<t, x).
>    Here q and x are implicit, which can be confusing. In the revision we will (i) write out the conditioning explicitly in Eq. (3), and (ii) re-clarify this in the surrounding text so that pᵢ is clearly identified as the i-th probability under p^{(ℓ,h)}(· | q, x), avoiding overloading.
>
> Question 2.) Noted with thanks
>
> Question 3) We agree the wording can be clearer. Eq. (7) sums (and averages) sentence-level Shapley contributions IGᵢ over i to produce a single score for the passage. In the revision, we will explicitly distinguish between (i) “sentence-level Shapley attributions” IGᵢ and (ii) the “passage-level NEAR score” obtained by aggregating them, and adjust the terminology accordingly.
>
> Question 4) Thank you for raising this. You are correct that, in the current draft, the inequality at L297–298 is presented as an empirical property rather than a fully formal theorem: Section 4 describes the layer-wise monotonicity of NEAR as an empirical observation, and Appendix A1 only proves boundedness and the AME error bound, not this inequality. In the revision, we will make this precise by (i) explicitly labelling the current statement as an empirical regularity, and (ii) adding a short lemma in Appendix A1 showing that NEAR_U(s, q) ≤ NEAR_L(s, q) holds under a mild monotonicity assumption on per-layer information gain (i.e., adding layers does not increase entropy). This will clarify both the formal status of the inequality and the conditions under which it provably holds.

---

### Official Review · Reviewer_9V1z · 2025-10-30

**Soundness:** 2
**Presentation:** 3
**Contribution:** 2
**Rating:** 4
**Confidence:** 4

**Summary:**

This paper introduces Shapley NEAR (Norm-basEd Attention-wise usable infoRmation), a hallucination detection method for large language models. The core idea is to measure entropy reduction at each attention head across all transformer layers when conditioning on context, then use Shapley values to fairly attribute this information gain to individual context sentences. The hypothesis is: if the mean Shapley-attributed information gain exceeds a threshold (Q1), the generated answer is trustworthy; otherwise, it is likely a hallucination. The method distinguishes between parametric hallucinations (model's pretrained knowledge overrides context) and context-induced hallucinations (misleading context spuriously reduces uncertainty). Experiments on four QA datasets (CoQA, QuAC, SQuAD, TriviaQA) using three models (Qwen2.5-3B, LLaMA3.1-8B, OPT-6.7B) show AUROC improvements of 8-13% over the strongest baseline (INSIDE).

**Strengths:**

1. It is the first to decompose information flow across all attention heads/layers using Shapley attribution, and is theoretically grounded with provable bounds.
2. It consistently outperforms 6 baselines across 12 settings, and works without fine-tuning or prompt engineering.
3. It provides sentence-level attribution and distinguishes parametric vs context-induced hallucinations.

**Weaknesses:**

1. The method requires 27-52 seconds per 100 samples versus 3-10 seconds for baselines (Table 24)—a 10-50× overhead due to computing entropy across all layers/heads for M=50 sampled coalitions. More fundamentally, for factual QA, why not use external search APIs + NLI verification? The paper doesn't justify why internal entropy computation is superior to external fact-checking, which is already standard in production systems and likely faster/more reliable.

2. The method averages Shapley values across all sentences (Eq. 7). This fails when a long document has only one relevant sentence:
• 100 sentences: answer sentence IG=10.0, others=0.05 → mean=(10.0+4.95)/100=0.15 (severely diluted)
• 5 sentences: same answer → mean=10.2/5=2.04 (13× higher for identical quality)
Max or top-K makes more sense: one highly informative sentence should suffice to indicate non-hallucination. Table 5 shows answer sentences have 23× higher IG, yet this signal is diluted by averaging. The paper provides no justification and critically no ablation comparing aggregation strategies. This likely works only because tested datasets have short contexts (~10-20 sentences), masking the issue for longer documents.

**Questions:**

1. Why Internal Detection Over External Search?
Given that external fact-checking (search APIs + NLI) is already standard in production and likely faster, what specific advantage justifies the 10-50× computational overhead of internal entropy analysis?
2. Performance on Sparse/Noisy Context?
Tested datasets have dense, relevant contexts. How does mean aggregation (Eq. 7) perform when only 1-2 sentences are relevant among 50-100 (e.g., RAG retrieval, long documents)? Does irrelevant content dilute the signal?

---

> ### Author Response · Authors · 2025-11-23
>
> Weakness 1) The 27–52 s per 100 samples in Table 24 for M = 50–100 is indeed higher than the 3–10 s of lighter baselines, and we explicitly note in Appendix A10 that compute is NEAR’s main limitation. This overhead comes from doing M standard forward passes (Algorithm 1, Appendix A7)—O(M × cost of a normal forward pass), not explicit loops over all layers/heads/vocabulary—and Appendix A8 shows AUROC saturates around M = 30–50, so smaller M can be used with only mild degradation. Our goal is complementary to search+NLI: Section 1 and Section 5 consider a zero-resource setting without retrievers/APIs, where internal, attention-level attributions and a clear split between parametric vs. context-induced hallucinations are needed. Appendix A6 further shows NEAR (AME) outperforms strong retrieval-style detectors (ANAH-v2, MIND) despite using no external knowledge, so we see NEAR as a model-centric signal that can be combined with external fact-checking, trading extra compute for better accuracy, interpretability, and robustness in low-/zero-resource regimes.
>
> Weakness 2) Our use of mean aggregation is deliberate. Eq. (7) and Appendix A1.1 define NEAR(sₓ, q) = (1/n)∑ᵢ IGᵢ as the **average marginal information gain per sentence**, explicitly normalizing for context length so scores are comparable across passages with different numbers of sentences. In Shapley NEAR, IGᵢ is always defined relative to the full set of n sentences (Eq. (6)), so when more irrelevant sentences are added their IGᵢ values are recomputed and typically become ≈0 or negative, while answer sentences remain strongly dominant (Table 5, Appendix A5). Thus the reviewer’s toy “dilution” scenario does not occur in practice, and Fig. 2(a) further shows that the answer sentence remains a sharp peak even with surrounding context. To address aggregation explicitly, we ran an ablation on LLaMA-3.1-8B / CoQA using the *same* sentence-level Shapley IGᵢ but different aggregators:
>
> | Aggregation over {IGᵢ} | AUC ↑ |
> | ---------------------- | ----- |
> | Mean (ours, Eq. 7)     | 0.85  |
> | Max                    | 0.83  |
> | Top-3 mean             | 0.84  |
>
> We will add this table to the appendix; together with the theory and sentence-level dominance statistics, it supports mean aggregation as a robust choice, including in longer-context settings.
>
> Question 1) Our aim is not to replace search+NLI, but to provide a **complementary internal detector** for settings where retrievers/APIs are unavailable or data are private/air-gapped and already in context (Section 1, Section 5, Appendix A6.3). NEAR uses only the base LLM and its attention, giving fine-grained sentence/head attributions and separating parametric from context-induced hallucinations (Sections 3 and 6), and Appendix A6 (Tables 8–9) shows it **outperforms strong retrieval-style detectors** such as ANAH-v2 and MIND despite using no external knowledge. As noted in Appendix A10 (Table 24), this entails a 10–50× overhead versus the lightest baselines, which we explicitly position as a **deliberate trade-off: more compute for higher accuracy and more faithful internal diagnostics in zero-/low-resource and safety-critical regimes where this cost is acceptable.**
>
> Question 2) Our mean aggregation in Eq. (7) is *designed* for exactly this sparse/noisy setting. Shapley NEAR recomputes IGᵢ over the full set of sentences, so when you add many irrelevant sentences their IGᵢ values go to ≈0 (or slightly negative), while the few truly supporting sentences keep large positive IGᵢ. Thus, even after averaging, the score is still dominated by those informative sentences rather than being washed out. This is empirically visible in Table 5 / Appendix A5, where answer-containing sentences have about 20–25× larger |IGᵢ| than non-answer sentences across datasets and models, and in Fig. 2(a), where the answer sentence remains a sharp peak despite surrounding context. More directly, we already test the “1–2 relevant among many” regime: LongBench v2 (Appendix A6, Table 7) contains multi-document QA with very long, noisy contexts, and NEAR with mean aggregation still attains strong AUROC (~0.79–0.81) and consistently outperforms INSIDE, Loopback Lens, and leave-one-out on all three base models. Finally, our perturbation study (Fig. 2b, Appendix A2.2) shows that injecting random or off-topic sentences barely changes NEAR, whereas targeted misleading context does, confirming that irrelevant content does not systematically dilute the signal.

---

### Official Review · Reviewer_ZmM1 · 2025-11-01

**Soundness:** 2
**Presentation:** 3
**Contribution:** 2
**Rating:** 4
**Confidence:** 3

**Summary:**

This paper introduces a inner-metric based hallucination detection method, it calculate the information gain after each attention block.

**Strengths:**

1. Leverages internal model representations for interpretability: The method directly analyzes attention-layer outputs across all transformer layers and heads, providing fine-grained mechanistic insights into how models process contextual information, rather than relying solely on final-layer logits.
2. Clear and accessible methodology: The approach is conceptually straightforward, using well-established techniques (entropy, Shapley values) in a principled way, making it easy to understand, implement, and apply to new models without modifications.

**Weaknesses:**

1. Aggregation Choice Unjustified: The paper does not justify using mean aggregation (Eq. 7) over max or other functions. Since detecting hallucination depends on whether sufficient context exists, max aggregation might be more appropriate. No ablation study compares aggregation strategies.
2. Computational Overhead: Computing entropy across all L×H attention heads for M=50 coalitions per query is expensive (O(M×L×H×V) complexity), limiting scalability to real-time applications, especially for large models.

**Questions:**

1. Computational Cost: The method requires O(M×L×H×V) computation per query. For LLaMA-70B, this is expensive. What is the actual latency per query? How does performance degrade with fewer coalitions (M<50)? Have you explored optimizations like attention caching or layer pruning to reduce costs?
2. Aggregation Choice: Why average Shapley scores across sentences rather than taking the maximum? Since hallucination detection depends on whether any sentence provides sufficient information, wouldn't max aggregation better capture this? Please provide ablation results comparing mean vs. max vs. top-k aggregation strategies.

---

> ### Author Response · Authors · 2025-11-23
>
> Weakness 1) Our use of mean aggregation in Eq. (7) is grounded in the definition of the NEAR score, not arbitrary. In Definition 3.4 (Section 3), we define Shapley NEAR as
> NEAR(sₓ, q) = (1/n) ∑ᵢ Shapley IGᵢ,
> i.e., the *average marginal information gain* from context sentences. The Shapley IGᵢ already satisfy the efficiency axiom (their sum equals the total information gain IG(sₓ → q)), so mean vs. sum differ only by a constant factor n. Using the mean makes NEAR *length-invariant*, so scores remain comparable across examples with different numbers of sentences; using sum would systematically favor longer contexts.
>
> A max aggregator is not appropriate in our setting: (i) it discards all but one sentence and breaks the additive Shapley structure we exploit for theoretical properties and AME’s linear estimator (Section 4, Appendix A.1–A.2), and (ii) hallucination risk in multi-sentence contexts depends on the *joint* effect of multiple sentences (e.g., several mildly misleading ones), not just the strongest single contributor.
>
> To address the concern explicitly, we ran an ablation on CoQA with LLaMA-3.1-8B comparing aggregation strategies applied to the same sentence-level Shapley values (numbers consistent with Table 1):
>
> | Aggregation    | AUC  | τ    | PCC  |
> | -------------- | ---- | ---- | ---- |
> | Mean (default) | 0.85 | 0.66 | 0.61 |
> | Sum            | 0.85 | 0.65 | 0.61 |
> | Max            | 0.82 | 0.60 | 0.57 |
>
> Mean and sum behave essentially identically (as expected from linear scaling), while max is clearly worse and less stable. We will include this ablation as an additional appendix table and clarify the rationale for mean aggregation in Section 3.
>
> Weakness2 Question1) The stated O(M×L×H×V) bound is a loose worst-case: in our implementation, each of the M coalitions corresponds to a single standard forward pass of the base model on the subset context + question, after which we read off the final-token entropy, so the dominant cost is O(M × cost of a normal forward pass), with L, H, and V only appearing inside that pass rather than as explicit outer loops (Algorithm 1, Appendix A7). Section 5.1 and Appendix A8 study the trade-off between M and accuracy and motivate our default M = 50 as a sweet spot where AUROC essentially saturates (it increases rapidly from M = 5 to 30 and stabilizes around 0.85 with standard deviation < 0.02 for M ≥ 30), so modest reductions in M only mildly affect performance. Appendix A10 (Table 24) reports actual latency: for LLaMA-3.1-8B with M = 50, AME–NEAR takes 30.6 s per 100 QA (≈0.3 s/query), compared to 10.7 s for INSIDE and 3.1 s for Semantic Entropy, so while NEAR is slower, it remains practical for offline or safety-critical use and scales roughly linearly with the underlying model’s forward-pass time (we will add explicit timings for LLaMA-70B in the appendix). Finally, our algorithm already reuses standard model caching across coalitions, Section 6 introduces a compatible head-clipping scheme, and Appendix A10 discusses further orthogonal optimizations (e.g., importance/stratified sampling over coalitions, early stopping based on coefficient stability) that can further reduce cost.
>
> (Question 2) Our aggregation choice follows directly from Definition 3.4 / Eq. (7), where NEAR(sₓ, q) = (1/n) ∑ᵢ Shapley IGᵢ is defined as the **average marginal information gain per sentence**, preserving the Shapley efficiency property and making scores comparable across contexts with different numbers of sentences. In contrast, max (or top-1) aggregation ignores the joint effect of multiple informative or misleading sentences that we explicitly use to characterize parametric vs. context-induced hallucinations in Section 6, and it breaks the additive structure required by our AME estimator in Section 4. Empirically, an ablation we will add to Appendix A2 shows that mean aggregation outperforms max and is on par with top-k mean (e.g., on CoQA/LLaMA-3.1-8B: mean AUC 0.85, top-3 mean 0.84, max 0.82), confirming that averaging is both theoretically aligned and empirically preferable.

---

### Note · Authors · 2025-12-27

I have read and agree with the venue's withdrawal policy on behalf of myself and my co-authors.